# Mitigating the Effect of Incidental Correlations on Part-based Learning

**Gaurav Bhatt**[*13], **Deepayan Das**[2], **Leonid Sigal**[13], **Vineeth N Balasubramanian** [2]

[1]The University of British Columbia,    [2]Indian Institute of Technology Hyderabad
[3] The Vector Institute, Canada

## Abstract

Intelligent systems possess a crucial characteristic of breaking complicated problems into smaller reusable components or parts and adjusting to new tasks using these part representations. However, current part-learners encounter difficulties in dealing with incidental correlations resulting from the limited observations of objects that may appear only in specific arrangements or with specific backgrounds. These incidental correlations may have a detrimental impact on the generalization and interpretability of learned part representations. This study asserts that part-based representations could be more interpretable and generalize better with limited data, employing two innovative regularization methods. The first regularization separates foreground and background information's generative process via a unique mixture-of-parts formulation. Structural constraints are imposed on the parts using a weakly-supervised loss, guaranteeing that the mixture-of-parts for foreground and background entails soft, object-agnostic masks. The second regularization assumes the form of a distillation loss, ensuring the invariance of the learned parts to the incidental background correlations. Furthermore, we incorporate sparse and orthogonal constraints to facilitate learning high-quality part representations. By reducing the impact of incidental background correlations on the learned parts, we exhibit state-of-the-art (SoTA) performance on few-shot learning tasks on benchmark datasets, including MiniImagenet, TieredImageNet, and FC100. We also demonstrate that the part-based representations acquired through our approach generalize better than existing techniques, even under domain shifts of the background and common data corruption on the ImageNet-9 dataset. The implementation is available on GitHub: `https://github.com/GauravBh1010tt/DPViT.git`

## 1 Introduction

Many datasets demonstrate a structural similarity by exhibiting "parts" or factors that reflect the underlying properties of the data [15, 18, 21, 28, 31, 43, 54]. Humans are efficient learners who represent objects based on their various traits or parts, such as a bird's morphology, color, and habitat characteristics. Part-based methods learn these explicit features from the data in addition to convolution and attention-based approaches (which only learn the internal representations), making them more expressive [7, 21, 41, 52, 54]. Most existing part-based methods focus on the unsupervised discovery of parts by modeling spatial configurations [14, 21, 52, 54, 61], while others use part localization supervision in terms of attribute vectors [25, 41, 50, 56] or bounding boxes [7]. Part-based methods come with a learnable part dictionary that provides a direct means of data abstraction and is effective in limited data scenarios, such as few-shot learning. Furthermore, the parts can be combined into hierarchical representations to form more significant components of the object

---

[*]**First author**; **Email**: gauravbhatt.cs.iitr@gmail.com

37th Conference on Neural Information Processing Systems (NeurIPS 2023).

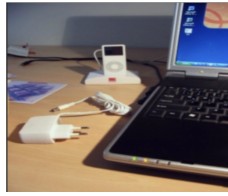 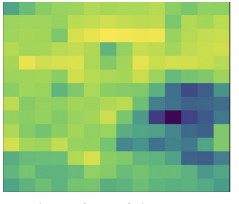 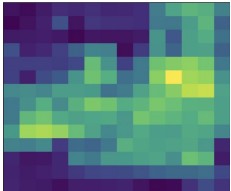

| (a) Input image | (b) ViT with parts | (c) Proposed - DPViT |

Figure 1: **Impact of incidental correlations on the interpretability of part learners.** We visualize the attention maps projected by the learned part dictionaries. Figure 1(b) illustrates the ViT-S backbone featuring a learnable part dictionary. However, it encounters difficulties in correctly identifying significant elements like the laptop, giving more attention to the background instead. In contrast, the proposed DPViT method successfully detects the most crucial parts of the image even in the presence of incidental correlations.

description [50, 55]. Part-based learning methods offer advantages in terms of interpretability and generalization, particularly in safety-critical domains like healthcare, aviation and aerospace industry, transportation systems (e.g., railways, highways), emergency response, and disaster management. These fields often face challenges in collecting large training samples, making part-based learning methods valuable. Furthermore, the ability to explain decisions becomes crucial in these contexts.

Various studies have indicated that correlations between image background and labels can introduce biases during machine learning model training [29, 32–34, 45, 46, 53, 59]. These correlations exist because specific background configurations create shortcuts for the model during training [32, 53]. While background information is crucial for decision-making, imbalanced background configurations can create unintended correlations, leading to undesirable outcomes. These correlations negatively impact the interpretability and generalization of part-based learners. For instance, let's consider a scenario where a laptop, a charger, and an iPod are on a table. In one case, let's examine the situation without any context or background information. Without background information, the model may struggle to understand the purpose and significance of components or parts such as the laptop, charger, and iPod on a table. It could fail to differentiate between these objects or grasp their functionalities, resulting in a lack of recognition and understanding. Conversely, suppose the model is predominantly trained with examples of these items on tables. In that case, it may overly focus on the background elements, such as the table itself, disregarding the individual entities. Thus it becomes essential to handle incidental correlations of image background to achieve a balanced understanding. Existing part-based approaches fail to handle incidental correlations that arise due to specific background signals dominating the training data (analogous to Figure 1(b)), thereby hampering their interpretability and generalization to limited data scenarios.

Having high-quality part representations is essential for achieving proficiency in part-based learning. In this context, quality pertains to the sparsity and diversity of the learned parts. Sparsity ensures that only a few parts are responsible for a given image, as images comprise a small subset of parts. Conversely, diversity in part representations prevents the parts from converging into a single representation and facilitates each part's learning of a unique data representation. Although incidental correlations can negatively affect learned parts' quality, the quality of part-based methods is a significant challenge that all part learners face. While previous studies have addressed the issue of part quality [50, 52], their solutions do not assure the sparsity and diversity of the learned parts, thereby failing to guarantee high-quality parts.

To solve the aforementioned challenges, we introduce the Disentangled Part-based Vision Transformer (DPViT), which is trained to be resilient to the incidental correlations of image backgrounds and ensures high-quality part representations. We propose a disentangled pre-training phase, which separates the generative process of the foreground and background information through a unique mixture-of-parts formulation. We impose structural constraints on the parts to ensure that the mixture-of-parts for foreground and background includes soft, object-agnostic masks without requiring direct supervision of part localization. The parts are learned to be invariant to incidental correlations using a self-supervised distillation fine-tune phase. To address the issue of the quality of learned parts, we impose sparse and spectral penalties on the part matrix to guarantee the high quality of learned part representations. We include an assessment of the sparse and spectral norms of the part matrix as a quantitative indicator of the learned parts' quality. Finally, we evaluate the effectiveness of our method on benchmark few-shot datasets, including MiniImagenet [35], TieredImageNet [40],

and FC100 [6]. To demonstrate the robustness of our proposed method to incidental correlations of backgrounds and common data corruptions, we use the benchmark ImageNet-9 dataset [53].

Our key contributions can be summarized as follows:

- We propose regularization techniques to disentangle the generative process of foreground and background information through a mixture-of-parts formulation. Additionally, we employ a self-supervised distillation regularization to ensure that the learned parts remain invariant to incidental correlations of the image background.
- We ensure the high quality of learned parts by employing both sparsity and spectral orthogonal constraints over the part matrix. These constraints prevent the parts from degenerating and encourage a diverse range of part representations.
- Apart from our evaluation of standard few-shot benchmark datasets, we also analyze the impact of incidental correlations of background and typical data distortions by utilizing the benchmark ImageNet-9 dataset [53].

## 2 Related Work

**Part-based learning**. The advantages of learning part-based representations have been extensively researched in image recognition tasks [15, 18, 31, 38, 43, 49, 52, 55, 56]. Earlier methods attempted to learn parts by defining a stochastic generative process [19, 43]. Part-based methods have been broadly classified into unsupervised and supervised categories. Unsupervised methods concentrate on learning the spatial relationship between parts by using part dictionaries without the supervision of part localization [14, 15, 20, 21, 28, 49, 52, 54, 61]. In contrast, supervised part-based methods rely on the supervision of part localization through attribute vectors [16, 27, 41, 50] or part bounding box information [7]. In the literature, parts are also referred to as concepts when supervision about part localization is involved [7, 41].

Discovering parts in an unsupervised way is a more challenging scenario that is applicable to most practical problems. Part dictionaries help data abstraction and are responsible for learning implicit and explicit data representations. For example, [28] clustered DCNN features using part-based dictionaries, while [26] introduced a generative dictionary-based model to learn parts from data. Similarly, [20] uses part-based dictionaries and an attention network to understand part representations. The ConstellationNet [54] and CORL [21] are some of the current methods from the constellation family [14, 61], and use dictionary-based part-prototypes for unsupervised discovery of parts from the data. Our approach also belongs to this category, as we only assume the part structure without requiring any supervision of part localization.

**Incidental correlations of image background**. Image backgrounds have been shown to affect a machine learning model's predictions, and at times the models learn by utilizing the incidental correlations between an image background and the class labels [4, 32, 42, 45, 46, 53, 58, 59]. To mitigate this issue, background researchers have used augmentation methods by altering background signals and using these samples during the training [32, 53, 58]. [53] performed an empirical study on the effect of image background on in-domain classification. They introduce several variants of background augmentations to reduce a model's reliance on the background. Similarly, [58] uses saliency maps of the image foreground to generate augmented samples to reduce the effect of the image background. Recently, [32] showed the effectiveness of background augmentation techniques for minimizing the effect of incidental correlations on few-shot learning.

Unlike these methods, our approach does not depend on background augmentations but instead learns the process of generating foreground and background parts that are disentangled. Furthermore, our proposed approach is not sensitive to the quality of foreground extraction and can operate with limited supervision of weak foreground masks.

**Few-shot learning and Vision Transformers**. In recent years, few-shot learning (FSL) has become the standard approach to evaluate machine learning models' generalization ability on limited data [1, 2, 5, 13, 24, 36, 44, 47, 54]. Vision transformers (ViT) [11] have demonstrated superior performance on FSL tasks [10, 22, 23, 30, 48, 51], and self-supervised distillation has emerged as a popular training strategy for these models [8, 17, 22, 30, 51]. A recent trend involves a two-stage procedure where models are pretrained via self-supervision before fine-tuning via supervised learning [8, 17, 22, 30, 60]. For example, [17] leverages self-supervised training with iBOT [60] as a pretext task, followed by inner loop token importance reweighting for supervised fine-tuning. HCTransformer [22] uses attribute surrogates learning and spectral tokens pooling for pre-training vision transformers

and performs fine-tuning using cascaded student-teacher distillation to improve data efficiency hierarchically. SMKD [30] uses iBOT pre-training and masked image modeling during fine-tuning to distill knowledge from the masked image regions.

Our approach employs a two-stage self-supervised training strategy of vision transformers akin to [22, 30, 60]. However, unlike existing methods that focus on generalization in few-shot learning, our primary objective is to learn part representations that are invariant to incidental correlations of the image background. Our training procedure is designed to facilitate learning disentangled and invariant part representations, which is impossible through existing two-stage self-supervised pipelines alone.

## 3 Problem Formulation and Preliminaries

In few-shot classification, the aim is to take a model trained on a dataset of samples from seen classes $\mathcal{D}^{seen}$ with abundant annotated data, and transfer/adopt this model to classify a set of samples from a disjoint set of unseen/novel classes $\mathcal{D}^{novel}$ with limited labeled data. Formally, let $\mathcal{D}^{seen} = \{(\mathbf{x}, y)\}$, where $\mathbf{x} \in \mathbb{X}$ corresponds to an image and $y \in \mathbb{Y}^{seen}$ corresponds to the label among the set of seen classes. We also assume that during training, we have limited supervision of class-agnostic foreground-background mask $(\mathcal{M}_f, \mathcal{M}_b)$ for regularization during training, which can be easily obtained by any weak foreground extractor as a preprocessing step (following [53]). Please note that no mask information is required for $\mathcal{D}^{novel}$ at inference.

We follow the work of [9, 60] on self-supervised training of ViTs to design our pretrain phase. During training, we apply random data augmentations to generate multiple views of the a given image $x^v \in \mathcal{D}^{seen}$. These views are then fed into both the teacher and student networks. Our student network, with parameters $\theta_s$, includes a ViT backbone encoder and a projection head $\phi_s$ that outputs a probability distribution over K classes. The ViT backbone generates a $[cls]$ token, which is then passed through the projection head. The teacher network, with parameters $\theta_t$, is updated using Exponentially Moving Average (EMA) and serves to distill its knowledge to the student by minimizing the cross-entropy loss over the categorical distributions produced by their respective projection heads.

$$\mathcal{L}_{cls} = \mathbb{E}_{(\mathbf{x}, y) \sim \mathcal{D}^{seen}} \mathcal{L}_{ce}(\mathcal{F}_\phi^t(\mathcal{F}_\theta^t(\mathbf{x^1})), \mathcal{F}_\phi^s(\mathcal{F}_\theta^s(\mathbf{x^2}))). \tag{1}$$

For inference, we use the standard $M$-way, $N$-shot classification by forming *tasks* ($\mathcal{T}$), each comprising of *support set* ($\mathcal{S}$) and *query set* ($\mathcal{Q}$), constructed from $\mathcal{D}^{novel}$. Specifically, a support set consists of $M \times N$ images; $N$ random images from each of $M$ classes randomly chosen from $\mathbb{Y}^{novel}$. The query set consists of a disjoint set of images, to be classified, from the same $M$ classes. Following the setup of [47], we form the class prototypes ($\mathbf{c}_m$) using samples from $\mathcal{S}$. The class prototypes and learned feature extractor ($\mathcal{F}_\theta$) are used to infer the class label $\hat{y}$ for an unseen sample $\mathbf{x}^q \in \mathcal{Q}$ using a distance metric $d$.

$$\hat{y} = \arg\max_m d(\mathcal{F}_\theta(\mathbf{x}^q), \mathbf{c}_m); \ \mathbf{c}_m = \frac{1}{N} \sum_{(\mathbf{x}, y_m) \in \mathcal{S}} \mathcal{F}_\theta(\mathbf{x}). \tag{2}$$

## 4 Proposed Methodology

Given an input sample $\mathbf{x} \in \mathbb{R}^{H \times W \times C}$, and a patch size $f$, we extract flattened 2D patches $\mathbf{x_f} \in \mathbb{R}^{N \times (F^2 \cdot C)}$, where $N$ is the number of patches generated and $(F, F)$ is the resolution of each image patch. Similar to a standard ViT architecture [11], we prepend a learnable $[class]$ token and positional embeddings to retain the positional information. The flattened patches are passed to multi-head self attention layers and MLP blocks to generate a feature vector $z_p = MSA(x_f)$.

Next, we define the parts as part-based dictionaries $\mathbf{P} = \{\mathbf{p}_k \in \mathbb{R}^{F^2 \cdot C}\}_{k=1}^K$, where $\mathbf{p}_k$ denotes the part-vector for the part indexed as $k$. The *part-matrix* ($\mathbf{P}$) is initialized randomly and is considered a trainable parameter of the architecture. Note that the dimension of each part-vector is equal to the dimension of flattened 2D patches, which is $F^2 \cdot C$. For each part $\mathbf{p}_k$, we compute a distance map $\mathbf{D}^k \in \mathbb{R}^N$ where each element in the distance map is computed by taking dot-product between the part $\mathbf{p}_k$ and all the $N$ patches: $\mathbf{D}^k = \mathbf{x_f} \cdot \mathbf{p}_k$.

Using the distance maps $\mathbf{D} \in \mathbb{R}^{N \times K}$, we introduce a multi-head cross-attention mechanism and compute the feature vector: $z_d = MCA(F_\psi(\mathbf{D}))$, where $F_\psi$ is an MLP layer which upsamples $\mathbf{D} : K \to F^2 \cdot C$. The cross-attention layer shares a similar design to self-attention layers; the only

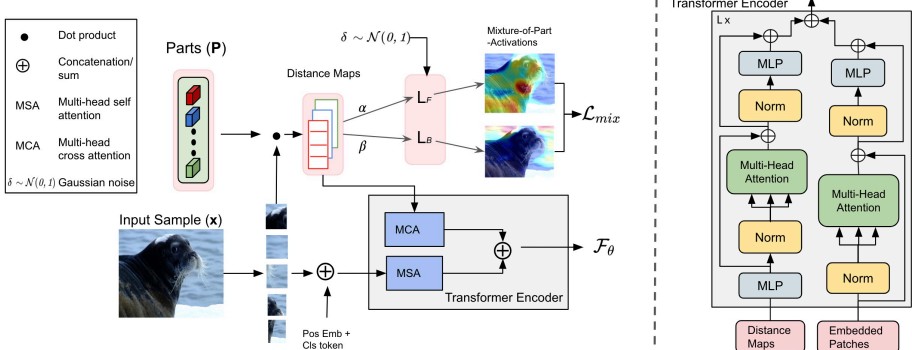

Figure 2: **Overview of proposed architecture - DPViT.** We employ a learnable part dictionary to generate a formulation incorporating foreground and background information. The spatial distance maps, computed by the part dictionary, are utilized to determine the mixture of latent codes for foreground and background. Our transformer encoder comprises multi-head self-attention (MSA) and multi-head cross-attention (MCA) layers. The MSA layer takes embedded patches as input, while the MCA layers utilize the distance maps as input.

difference is the dimensions of input distance maps. The cross-attention helps contextualize information across part-dictionary and the spatial image regions and provides complementary properties to $MSA$ layers. (Please refer to Appendix for experiments on complementary properties of $MSA$ and $MCA$).

Finally, we add the output feature vectors of $MSA$ and $MCA$ to form the feature extractor $\mathcal{F}_\theta$ defined in Eqn 1:

$$\mathcal{F}_\theta = [z_p \oplus z_d] \tag{3}$$

**Disentanglement of foreground-background space using mixture-of-parts**. We start by dividing the parts-matrix $\mathbf{P} \in \mathbb{R}^{K \times F^2 \cdot C}$ into two disjoint sets: foreground set $\mathbf{P_F} \in \mathbb{R}^{n_f \times C}$ and background set $\mathbf{P_B} \in \mathbb{R}^{n_b \times C}$, such that $K = n_f + n_b$.

Next, we construct latent variables to aggregate the foreground and background information using a mixture-of-parts formulation over the computed distance maps $\mathbf{D}$:

$$L_F = \sum_{k \in n_f} \alpha_k \mathbf{D}^k + \delta_f; L_B = \sum_{k \in n_b} \beta_k \mathbf{D}^k + \delta_b \tag{4}$$

where, $\alpha_k$ and $\beta_k$ are the learnable weights given to the $k^{th}$ part-vector in the corresponding mixture, whereas $\delta_f$ and $\delta_b$ are Gaussian noises sampled from $\mathcal{N}(0,1)$. The Gaussian noise is added to the latent codes to ensure that mixture-of-parts are robust to common data distortions. Please note that the purpose of Gaussian noise is not to induce variability in the latent codes, as foreground information for a given image is deterministic.

Finally, our disentanglement regularization takes the form of an alignment loss between the latent codes and the class-agnostic foreground-background masks:

$$\mathcal{L}_{mix} = ||\mathcal{I}(L_F) - \mathcal{M}_f||_2 + ||\mathcal{I}(L_B) - \mathcal{M}_b||_2 \tag{5}$$

where, $\mathcal{I}(L)$ is the bilinear interpolation of a given latent code $L$ to the same size as $\mathcal{M}$.

During the architectural design phase, we employ distinct components ($\mathbf{P}$) for each encoder block. Consequently, $z_p$ and $z_d$ are calculated in an iterative manner and subsequently transmitted to the subsequent encoder block. Regarding the computation of $L_{mix}$, we utilize the distance maps $\mathbf{D}$ from the concluding encoder block. While it's feasible to calculate $L_{mix}$ iteratively for each block and then aggregate them for a final $L_{mix}$ computation, our observations indicate that this approach amplifies computational expenses and results in performance deterioration. As a result, we opt to compute $L_{mix}$ using the ultimate encoder block.

**Learning high-quality part representations**. A problem with minimizing the mixture objective defined in Eqn 5 is that it may cause the degeneration of parts, thereby making the part representations less diverse. One solution is to enforce orthogonality on the matrix $\mathbf{P}^{m \times n}$ by minimizing $||\mathbf{P}^T \mathbf{P} - \mathbf{I}||$, similar to [50]. However, the solution will result in a biased estimate as $m < n$; that is, the number of

parts ($K$) is always less than the dimensionality of parts ($F^2 \cdot C$). In our experiments, we observed that increasing $K$ beyond a certain threshold degrades the performance as computational complexity increases, and is consistent with the findings in [54]. (Please refer to our Appendix section for experiments on the different values of $K$). To minimize the degeneration of parts, we design our quality assurance regularization by minimizing the spectral norm of $\mathbf{P}^T\mathbf{P} - \mathbf{I}$, and by adding $L_1$ sparse penalty on the part-matrix $\mathbf{P}$. The spectral norm of $\mathbf{P}^T\mathbf{P} - \mathbf{I}$ has been shown to work with over-complete ($m < n$) and under-complete matrices ($m \geq n$) [3].

$$\mathcal{L}_Q(\lambda_s, \lambda_o) = \lambda_s ||\mathbf{P}||_1 + \lambda_o \Big[ \sigma\big(\mathbf{P_F} \cdot \mathbf{P_F}^T - \mathbf{I}\big) + \sigma\big(\mathbf{P_B} \cdot \mathbf{P_B}^T - \mathbf{I}\big) \Big] \tag{6}$$

where $\mathbf{I}$ is the identity matrix, $\lambda_s$ and $\lambda_o$ are the regularization coefficients for sparsity and orthogonality constraints. $\sigma(\mathbf{P})$ is the spectral norm of the matrix $\mathbf{P}$ which is computed using the scalable power iterative method described in [3].

**Disentangled Pretraining Objective**. We pretrain DPViT using the following loss function:

$$\mathcal{L}_{PT} = \lambda_{cls}\mathcal{L}_{cls} + \lambda_{mix}\mathcal{L}_{mix} + \mathcal{L}_Q(\lambda_s, \lambda_o) \tag{7}$$

where $\lambda_{cls}, \lambda_{mix}, \lambda_s$, and $\lambda_o$ are the weights given to each loss term and are tuned on the validation set.

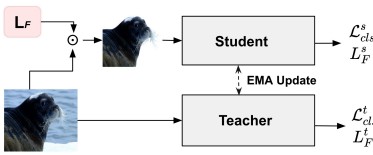

Figure 3: Invariant fine-tuning of DPViT via distillation framework.

### 4.1 Invariant fine-tuning

In the pretrain phase, our approach learns part representations that are disentangled and diverse, but it does not achieve invariance to the incidental correlations of image background. During the fine-tuning stage, we utilize the learned foreground latent code to extract the relevant foreground information from a given image $x$: $x_f = x \odot \mathcal{I}(L_F)$, where $\odot$ denotes the Hadamard product. The teacher network receives the original image, while the student network receives the foreground-only image. By distilling knowledge between the $[class]$ tokens and foreground latent codes $L_F$ of the student and teacher networks, we achieve invariance to the incidental correlations of image background.

$$\mathcal{L}_{cls}^{inv} = \mathcal{L}_{ce}(\mathcal{F}_\phi^t(\mathcal{F}_\theta^t(x)), \mathcal{F}_\phi^s(\mathcal{F}_\theta^s(x_f))); \mathcal{L}_p^{inv} = \mathcal{L}_{ce}(L_F^t(x), L_F^s(x_f)) \tag{8}$$

The two proposed invariant regularizations serve distinct purposes: $\mathcal{L}_{cls}^{inv}$ encourages the model to classify images independently of the background, while $\mathcal{L}_p^{inv}$ ensures that the latent foreground code captures relevant foreground information even when the background is absent, making the learned parts invariant to the incidental correlations.

**Invariant Fine-tuning Objective**. Finally, our fine-tuning objective is given as :

$$\mathcal{L}_{FT} = \lambda_{cls}\mathcal{L}_{cls} + \lambda_{cls}^{inv}\mathcal{L}_{cls}^{inv} + \lambda_p^{inv}\mathcal{L}_p^{inv} \tag{9}$$

where $\lambda_{cls}, \lambda_{cls}^{inv}$, and $\lambda_p^{inv}$ are the weights given to each loss term and are tuned on the validation set after pretraining.

## 5 Experiments

We evaluate the proposed approach on four datasets: MiniImageNet [35], TieredImageNet [40], FC100 [37], and ImageNet-9 [53]. The MiniImageNet, TieredImageNet, and FC100 are generally used as benchmark datasets for few-shot learning. For MiniImageNet, we use the data split proposed in [39], where the data samples are split into 64, 16, and 20 for training, validation, and testing, respectively. The TieredImageNet [40] contains 608 classes divided into 351, 97, and 160 for meta-training, meta-validation, and meta-testing. On the other hand, FC100 [37] is a smaller resolution dataset ($32 \times 32$) that contains 100 classes with class split as 60, 20, and 20.

To investigate the impact of background signals and data corruption on classifier performance, researchers introduced ImageNet-9 (IN-9L) [53]. IN-9L is a subset of ImageNet comprising nine coarse-grained classes: dog, bird, vehicle, reptile, carnivore, insect, instrument, primate, and fish. Within these super-classes, there are 370 fine-grained classes, with a training set of 183,006 samples. The authors of [53] created different test splits by modifying background signals, resulting in 4050

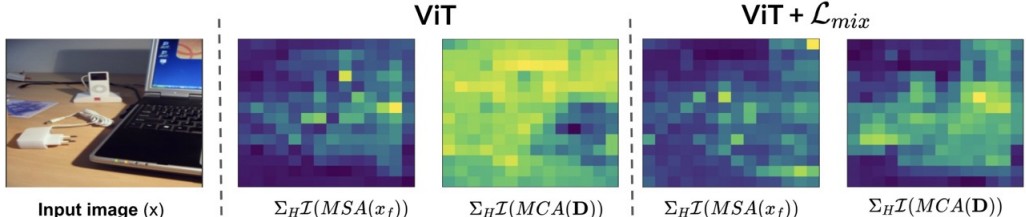

**ViT**       **ViT + $\mathcal{L}_{mix}$**

**Input image (x)**    $\Sigma_H \mathcal{I}(MSA(x_f))$    $\Sigma_H \mathcal{I}(MCA(\mathbf{D}))$    $\Sigma_H \mathcal{I}(MSA(x_f))$    $\Sigma_H \mathcal{I}(MCA(\mathbf{D}))$

Figure 4: Visualizing *MSA* and *MCA* layers. The joint representation is obtained by averaging all attention heads ($\sum_H$). We study the effect of $\mathcal{L}_{mix}$ on the interpretability of part based learners.

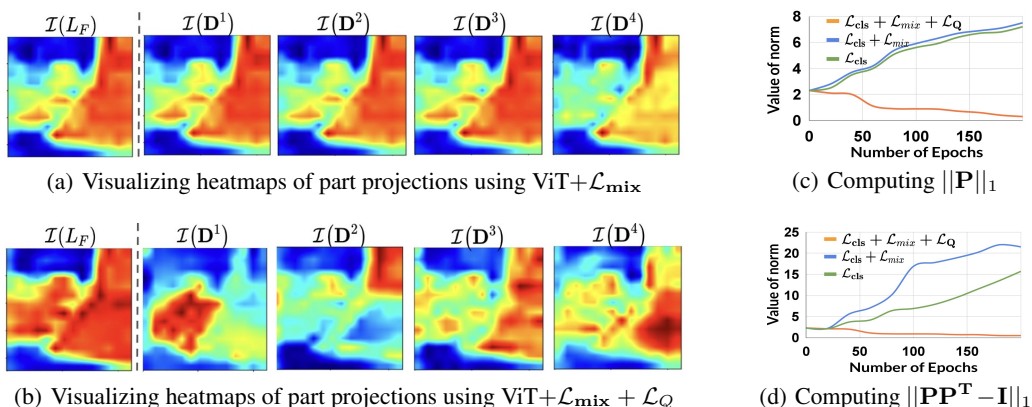

(a) Visualizing heatmaps of part projections using ViT+$\mathcal{L}_{\mathbf{mix}}$

(c) Computing $||\mathbf{P}||_1$

(b) Visualizing heatmaps of part projections using ViT+$\mathcal{L}_{\mathbf{mix}} + \mathcal{L}_Q$

(d) Computing $||\mathbf{PP^T} - \mathbf{I}||_1$

Figure 5: **Visualizing the quality of learned parts for input image from Figure 4.** In Figure 5(a) and 5(b), we show the foreground mixture $\mathcal{I}(L_F)$ and four foreground parts selected randomly from $n_f$. Meanwhile, Figure 5(c) and 5(d) show the sparse and orthogonal norms as metrics for evaluating the quality of the part matrix $\mathbf{P}$.

samples per split to evaluate various classifiers. We use three of these test splits for our evaluation: Original (with no changes to the background), M-SAME (altering the background from the same class), and M-RAND (altering the background from a different class). Additionally, [53] introduced a metric called BG-GAP to assess the tendency of classifiers to rely on background signal, which is measured as the difference in performance between M-SAME and M-RAND.

We use a ViT-S backbone for all our experiments and follow the same pipeline in iBOT [60] for pre-training, keeping most hyperparameters unchanged. We use a batch size of 480 with a learning rate of 0.0005 decayed alongside a cosine schedule and pre-train DPViT for 500 epochs for all four datasets. Fine-tuning is carried out on the same train data for 50 epochs utilizing the class labels (similar to [22]). The DPViT architecture has $K$=64 and $n_f$=40 for all of our experiments. The pre-training and fine-tuning are carried out on 4 A40 GPUs. More details related to the architectures and optimization are provided in Appendix.

We start our experimental study by examining how incidental correlation affects part-learners' interpretability and quality of learned parts. This study is conducted on the MiniImageNet dataset. We then present experimental results on few-shot learning on the MiniImageNet, TieredImageNet, and FC100 datasets. Lastly, we provide a quantitative analysis of the influence of incidental background correlations on the various test splits in the ImageNet-9 dataset, followed by ablation studies and a discussion.

### 5.1 How do incidental correlations affect the interpretability of part learners?

To examine how incidental correlations impact various components, we solely employ the $\mathcal{L}_{cls}$ loss to train DPViT on the MiniImageNet dataset. (Note that $\mathcal{L}_{cls}$ is equivalent to ViT with parts). We then present a visualization of the attention layers ($MSA$ and $MCA$) in Figure 4. The $MSA$ layers effectively recognize relevant in-context information of some objects, but the $CSA$ layers fail to pinpoint foreground details. This is because incidental correlations in the background dominate the $CSA$ layers. However, incorporating the $\mathcal{L}_{mix}$ regularization results in improved localization by the $MCA$ layers, which are no longer influenced by incidental correlations. An important point to highlight is that the design of $MSA$ layers causes them to focus on objects specific to a particular class. As a result, their effectiveness is reduced when multiple objects are present (as seen in Figure 4, where MSA misses objects on the left side). Conversely, $CSA$ layers learn class-agnostic features

| Method | Backbone | MiniImageNet | | TieredImageNet | | FC100 | |
|---|---|---|---|---|---|---|---|
| | | 1-shot | 5-shot | 1-shot | 5-shot | 1-shot | 5-shot |
| **ProtoNets** (2017) [47] | ResNet12 | $60.39_{\pm0.16}$ | $78.53_{\pm0.25}$ | $65.65_{\pm0.92}$ | $83.40_{\pm0.65}$ | $37.50_{\pm0.60}$ | $52.50_{\pm0.60}$ |
| **DeepEMD v2** (2020) [57] | ResNet12 | $68.77_{\pm0.29}$ | $84.13_{\pm0.53}$ | $71.16_{\pm0.87}$ | $86.03_{\pm0.58}$ | $46.47_{\pm0.26}$ | $63.22_{\pm0.71}$ |
| **COSOC** (2021) [32] | ResNet12 | $69.28_{\pm0.49}$ | $85.16_{\pm0.42}$ | $73.57_{\pm0.43}$ | $87.57_{\pm0.10}$ | - | - |
| **MixtFSL** (2021) [1] | ResNet12 | $63.98_{\pm0.79}$ | $82.04_{\pm0.49}$ | $70.97_{\pm1.03}$ | $86.16_{\pm0.67}$ | - | - |
| **Match-feat** (2022) [2] | ResNet12 | $68.32_{\pm0.62}$ | $82.71_{\pm0.46}$ | $71.22_{\pm0.86}$ | $85.43_{\pm0.55}$ | - | - |
| **Label-Halluc** (2022) [24] | ResNet12 | $67.04_{\pm0.70}$ | $85.87_{\pm0.48}$ | $71.97_{\pm0.89}$ | $86.80_{\pm0.58}$ | $47.37_{\pm0.70}$ | $67.92_{\pm0.70}$ |
| **FeLMi** (2022) [44] | ResNet12 | $67.47_{\pm0.78}$ | $86.08_{\pm0.44}$ | $71.63_{\pm0.89}$ | $87.07_{\pm0.55}$ | $49.02_{\pm0.70}$ | $68.68_{\pm0.70}$ |
| **SUN** (2022) [10] | VIT | $67.80_{\pm0.45}$ | $83.25_{\pm0.30}$ | $72.99_{\pm0.50}$ | $86.74_{\pm0.33}$ | - | - |
| **FewTure** (2022) [23] | Swin-Tiny | $72.40_{\pm0.78}$ | $86.38_{\pm0.49}$ | $76.32_{\pm0.87}$ | $89.96_{\pm0.55}$ | $47.68_{\pm0.78}$ | $63.81_{\pm0.75}$ |
| **HCTransformer** (2022) [22] | 3× VIT-S | **$74.74_{\pm0.17}$** | $89.19_{\pm0.13}$ | **$79.67_{\pm0.20}$** | $91.72_{\pm0.11}$ | $48.27_{\pm0.15}$ | $66.42_{\pm0.16}$ |
| **SMKD** (2023) [30] | VIT-S | $74.28_{\pm0.18}$ | $88.82_{\pm0.09}$ | $78.83_{\pm0.20}$ | $91.02_{\pm0.12}$ | $50.38_{\pm0.16}$ | $68.37_{\pm0.16}$ |
| **ConstNet** (2021) [54] | ResNet12 | $64.89_{\pm0.23}$ | $79.95_{\pm0.17}$ | $70.15_{\pm0.76}$ | $86.10_{\pm0.70}$ | $43.80_{\pm0.20}$ | $59.70_{\pm0.20}$ |
| **TPMN** (2021) [52] | ResNet12 | $67.64_{\pm0.63}$ | $83.44_{\pm0.43}$ | $72.24_{\pm0.70}$ | $86.55_{\pm0.63}$ | $46.93_{\pm0.71}$ | $63.26_{\pm0.74}$ |
| **CORL** (2023) [21] | ResNet12 | $65.74_{\pm0.53}$ | $83.03_{\pm0.33}$ | $73.82_{\pm0.58}$ | $86.76_{\pm0.52}$ | $44.82_{\pm0.73}$ | $61.31_{\pm0.54}$ |
| **VIT-with-parts** ($L_{cls}$) | VIT-S | $72.15_{\pm0.20}$ | $87.61_{\pm0.15}$ | $78.03_{\pm0.19}$ | $89.08_{\pm0.19}$ | $48.92_{\pm0.13}$ | $67.75_{\pm0.15}$ |
| **Ours - DPViT** | VIT-S | $73.81_{\pm0.45}$ | **$89.85_{\pm0.35}$** | $79.32_{\pm0.48}$ | **$91.92_{\pm0.40}$** | **$50.75_{\pm0.23}$** | **$68.80_{\pm0.45}$** |

Table 1: Evaluating the performance of our proposed method on three benchmark datasets for few-shot learning - MiniImageNet, Tiered-ImageNet, and FC100. The top blocks show the non-part methods while the bottom block shows the part-based methods. The best results are bold, and ± is the 95% confidence interval in 600 episodes.

| Method | IN-9L ↑ | Original ↑ | M-SAME ↑ | M-RAND ↑ | BG-GAP ↓ |
|---|---|---|---|---|---|
| *ResNet-50* [53] | 94.6 | 96.3 | 89.9 | 75.6 | 14.3 |
| *WRN-50×2* [53] | 95.2 | 97.2 | 90.6 | 78.0 | 12.6 |
| *ConstNet* | 90.6 | 92.7 | 86.1 | 69.2 | 17.1 |
| *ViT-S pre* [11] | 82.5 | 84.9 | 72.2 | 50.3 | 21.9 |
| *CT* [41] | 84.7 | 85.5 | 73.1 | 51.5 | 21.6 |
| *VIT-with-parts* | 95.1 | 97.2 | 91.5 | 81.7 | 9.8 |
| **Ours - DPViT** | 96.9 | 98.5 | 93.4 | 87.5 | 5.9 |

Table 2: Performance evaluation on domain shift of varying background and common data corruptions on ImageNet-9. Evaluation metric is Accuracy %.

| Method | 1-shot ↑ | 5-shot ↑ | $\|\mathbf{P}\|_1 \downarrow$ | $\|\mathbf{PP}^T - \mathbf{I}\|_1 \downarrow$ |
|---|---|---|---|---|
| SMKD [30] | 60.93 | 80.38 | - | - |
| $\mathcal{L}_{cls}$ | 61.24 | 81.12 | 8.41 | 25.82 |
| $\mathcal{L}_{cls} + \mathcal{L}_{mix}$ | 62.81 | 83.25 | 8.73 | 24.61 |
| $\mathcal{L}_{cls} + \mathcal{L}_{mix} + \mathcal{L}_Q$ | 62.15 | 82.95 | 0.35 | 0.56 |

Table 3: Ablation of different loss terms during pre-training of DP-ViT. We show the effect of each loss term on the MiniImageNet dataset.

consistently throughout the training set, enabling them to perform well in the presence of multiple objects. For further investigation of the complementary properties of $MSA$ and $MCA$, please see the Appendix. Additionally, the Appendix includes visualizations of individual attention heads.

## 5.2 Studying the quality of learned part representations

As previously stated, the interpretability of learned parts is directly influenced by the sparsity and diversity of the part matrix $\mathbf{P}$. This is achieved by examining the impact of $\mathcal{L}_Q$ during pretraining. We present visualizations of the learned foreground parts in Figure 5(a) and 5(b) for the input image $x$ from Figure 4. While both setups successfully learn the foreground mixture, $\mathcal{I}(L_F)$, without $\mathcal{L}_Q$, the parts degenerate into a homogeneous solution (Figure 5(a)), lacking sparsity and diversity. With the inclusion of $\mathcal{L}_Q$, however, the learned parts become sparse and diverse (Figure 5(b)).

We use the $L$-1 and orthogonal norms of the part matrix to assess the sparsity and diversity of the parts. As illustrated in Figure 5(c) and Figure 5(d), the addition of $\mathcal{L}_Q$ maintains bounded norms, inducing sparsity and diversity among the parts. The results also highlight higher norm values by employing $\mathcal{L}_{cls} + \mathcal{L}_{mix}$ losses, which show the degeneracy caused by the introduction of mixture loss. Moreover, the higher sparse and orthogonal norms for $\mathcal{L}_{cls}$ demonstrate that part-based methods generally do not maintain the quality of parts.

## 5.3 Few-shot Learning

We compare DPViT with recent part-based methods: ConstNet [54], TPMN [52], and CORL [21]; ViT-based methods: SUN [10], FewTure [23], HCTransformer [22], and SMKD [30] (for SMKD we compare with their prototype-based few-shot evaluation on all the datasets); and recent ResNet based few-shot methods: *Match-feat* [2], *Label-halluc* [24], and *FeLMi* [44].

As shown in Table 1, the proposed method outperforms existing part-based methods by clear margins. Moreover, DPViT achieves competitive performance compared to current ViT-based methods, especially on the FC100 dataset with low-resolution images. The FSL results show that the self-supervised ViT methods give an edge over the existing ResNet-based backbones.

| Setting | Org ↑ | M-S ↑ | M-R ↑ | $\|\mathbf{P}\|_1 \downarrow$ | $\|\mathbf{P}\mathbf{P}^T - \mathbf{I}\|_1 \downarrow$ |
|---|---|---|---|---|---|
| $\mathcal{L}_{cls}$ | 98.1 | 91.5 | 82.7 | 1.4 | 2.3 |
| $\mathcal{L}_{cls} + \mathcal{L}_{cls}^{inv}$ | 98.5 | 93.4 | 87.5 | 1.4 | 2.3 |
| $\mathcal{L}_{cls} + \mathcal{L}_{cls}^{inv} + \mathcal{L}_{p}^{inv}$ | 98.2 | 93.1 | 87.2 | 0.3 | 0.5 |

Table 4: Ablation of different loss terms during fine-tuning of DP-ViT. We show the effect of each loss term on the ImageNet-9 test splits.

| Method | Backbone | MiniImageNet -> CUB |
|---|---|---|
| **ProtoNet** | ResNet-18 | 67.1 |
| **MixtFSL** [2] | ResNet-18 | 68.7 |
| **ConstNet** [54] | ResNet-12 | 68.8 |
| **SUN** [10] | ViT-S | 72.1 |
| **Proposed-DPViT** | ViT-S | **77.1** |

Table 5: Cross-domain evaluation. We evaluate models trained on MiniImageNet on the CUB dataset.

| Setting | 1-shot ↑ | 5-shot ↑ | $\|\mathbf{P}\|_1 \downarrow$ | $\|\mathbf{P}\mathbf{P}^T - \mathbf{I}\|_1 \downarrow$ |
|---|---|---|---|---|
| $\lambda_s=0.1, \lambda_o=0.1$ | 73.9 | 89.9 | 0.7 | 0.9 |
| $\lambda_s=0.5, \lambda_o=0.5$ | 73.8 | 89.8 | 0.3 | 0.5 |
| $\lambda_s=2.0, \lambda_o=2.0$ | 72.4 | 88.2 | 0.1 | 0.2 |

Table 6: Evaluating the tradeoff between interpretability and few-shot generalization on MiniImageNet.

| | 0% | | 10% | | 50% | | 100% | |
|---|---|---|---|---|---|---|---|---|
| | 1-s | 5-s | 1-s | 5-s | 1-s | 5-s | 1-s | 5-s |
| DPViT | 72.3 | 88.1 | 72.9 | 88.6 | 73.7 | 89.8 | 73.8 | 89.8 |

Table 7: Results on limited supervision of foreground masks on the MiniImageNet dataset. Here, we use 1-s for 1-shot and 5-s for 5-shot accuracy.

### 5.4 Studying the impact of incidental correlations of background on ImageNet-9

The impact of incidental background correlation on the ImageNet-9 dataset is examined, and the corresponding results are presented in Table 8. DPViT achieves the highest performance across IN-9L, Original, M-SAME, and M-RAND among the different test splits. Remarkably, our proposed DPViT method significantly reduces the BG-GAP value to $5.9$, indicating its resilience to the effects of incidental correlation caused by varying backgrounds. In comparison to non-part methods such as ResNet-50 and WRN-50 $\times$ 2 (as shown in Table 8), ConstNet [54], which is a part-based method, suffers more from the negative impact of incidental correlation, underscoring the detrimental effect on the generalization of part-learners. To evaluate ConstNet, we utilize the provided source code by the authors to train their model on the IN-9L training data.

## 6 Ablation study and Discussion

In Section 5.1 and 5.2, we investigate how the inclusion of $\mathcal{L}_{mix}$ and $\mathcal{L}_Q$ impacts the interpretability of DPViT during the pretraining phase. We will now conduct an ablation study to assess the impact of both these loss components during the pre-training phase. As indicated in Table 3, introducing $\mathcal{L}_{mix}$ results in enhanced 1-shot and 5-shot performance compared to the baseline model represented by $\mathcal{L}_{cls}$. It's worth noting that the SMKD [30] model performs worse than the part-ViT baseline ($\mathcal{L}_{cls}$). Upon incorporating $\mathcal{L}_Q$, the few-shot performance experiences a slight decrease, but the norms remain constrained, indicating improved quality of part representations.

Additionally, in this section, we examine the advantages of incorporating $\mathcal{L}_{cls}^{inv}$ and $\mathcal{L}_{p}^{inv}$ during the fine-tuning process on the ImageNet-9 dataset. This ablation study focuses on the significance of invariance terms in handling background-related incidental correlations. The results presented in Table 4 demonstrate that the introduction of $\mathcal{L}_{cls}^{inv}$ enhances generalization on M-S (M-SAME) and M-R (M-RAND), thereby illustrating the benefits of inducing invariance through $\mathcal{L}_{cls}^{inv}$ across varying backgrounds. Moreover, the inclusion of $\mathcal{L}_{cls}^{inv}$ yields a lower BG-GAP value of $5.9$, in contrast to $8.8$ obtained without $\mathcal{L}_{cls}^{inv}$. Notably, the introduction of $\mathcal{L}_{cls}^{inv}$ has no impact on the quality of parts, as evidenced by the final norm values of $\mathbf{P}$. By employing $\mathcal{L}_{p}^{inv}$ in conjunction with other losses, the norms remain bounded, thereby preserving the interpretability of parts. We also observe a minimal decrease in classification performance upon introducing $\mathcal{L}_{p}^{inv}$, highlighting the trade-off between generalization and interpretability that will be explored in the following section.

### 6.1 Cross-domain evaluation

In order to assess the advantages of eliminating incidental correlations for transfer learning, we perform experiments on a cross-domain dataset, specifically MiniImageNet to CUB. In this setup, we evaluate the performance of models trained on MiniImageNet when applied to the CUB dataset. As demonstrated in Table 5, the DPViT model stands out by achieving the highest classification performance. These experiments highlight the clear benefits of employing part-based methods and eliminating incidental correlations.

### 6.2 Tradeoff between interpretability and generalization

We observe a tradeoff between the interpretability of parts and their generalization. To explore this, we conduct an ablation study during pre-training on the MiniImageNet dataset. As indicated in Table

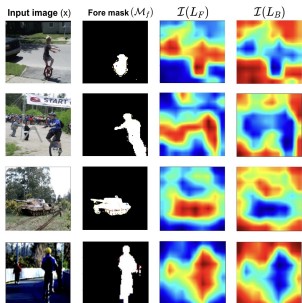
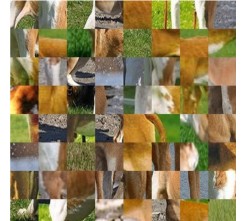 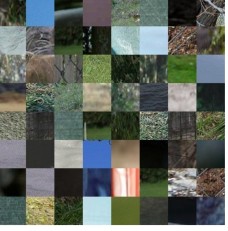

(a) Foreground patches    (b) Background patches

Figure 6: Robustness of DPViT towards a weak foreground extractor on MiniImageNet.

Figure 7: Disentanglement of foreground/background patches extracted by DPViT.

6, assigning higher values to $\lambda_s$ and $\lambda_o$ places greater emphasis on $\mathcal{L}_Q$, leading to lower norm values and consequently improving the quality of parts. However, this also results in a slight reduction in few-shot accuracy. After careful analysis, we determine that an optimal value of $0.5$ for both $\lambda_s$ and $\lambda_o$ strikes a balance, maintaining the quality of parts while preserving few-shot generalization.

### 6.3 Partial observability of foreground mask

Our training procedure employs foreground masks acquired through a foreground extractor, similar to the one described in [53]. To study the dependence of DPViT on the availability of foreground masks, we examine the weak/limited supervision scenario for foreground masks, where only a small subset of samples possesses the corresponding masks. As depicted in Table 7, we observe that DPViT achieves comparable performance even when only 10% of the training samples have mask information. The performance difference is less than 1.5% for 5-shot and less than 0.8% for 1-shot performance. Furthermore, Figure 7 presents visualizations of image patches surrounding a random foreground and a background part. We also find that in the setup with a $0\%$ foreground mask, equivalent to $\lambda_{mix} = 0$, no disentanglement is observed in the extracted patches. (More visualizations can be found in the Appendix section).

### 6.4 Working with weak foreground extractor

The features of DPViT ($\mathcal{F}_\theta$) depend entirely on the input sample $\mathbf{x}$ in order to learn part representations, without explicitly incorporating the mask information. The mask serves as a weak signal to regularize our training objective and separate the foreground parts from the background. Additionally, introducing Gaussian noise in the latent codes enhances DPViT's ability to handle misalignment issues with the foreground masks. Consequently, the learned features remain unaffected by mistakes made by the existing foreground extractor. Moreover, we find that the mixture-of-parts representations can accurately determine the foreground location even when the mask information is missing or incorrect (as illustrated in Figure 6).

### 6.5 Limitations of DPViT

A constraint within our framework involves relying on a pre-existing foreground extractor. In certain scenarios, such as the classification of tissue lesions for microbiology disease diagnosis, obtaining an existing foreground extractor might not be feasible. Similarly, DPViT focuses on learning components that are connected to the data, yet it doesn't encompass the connections between these components, like their arrangement and hierarchical combination. Introducing compositional relationships among these components could enhance comprehensibility and facilitate the creation of a part-based model capable of learning relationships among the parts.

## 7 Conclusion

In this work, we study the impact of incidental correlations of image backgrounds on the interpretability and generalization capabilities of part learners. We introduce DPViT, a method that effectively learns disentangled part representations through a mixture-of-parts approach. Furthermore, we enhance the quality of part representations by incorporating sparse and orthogonal regularization constraints. Through comprehensive experiments, we demonstrate that DPViT achieves competitive performance comparable to state-of-the-art methods, all while preserving both implicit and explicit interpretability.

## Acknowledgments and Disclosure of Funding

The computation resources used in preparing this research were provided by Digital Research Alliance of Canada, and The Vector Institute, Canada.

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

# Mitigating the Effect of Incidental Correlations on Part-based Learning (Supplementary Material)

## A  Appendix

### A.1  Why the term "incidental correlations" for image background?

The concept of "incidental correlations" is derived from the notion of incidental endogeneity [12], which describes unintentional but genuine correlations between variables. In the context of our study, image backgrounds are not considered spurious because they offer contextual information that aids in decision-making. Therefore, the relationship between image backgrounds and classification is not anti-causal, as would be true if the backgrounds were spurious. We argue that the imbalance of specific image backgrounds in the training data is the primary factor contributing to the introduction of incidental correlations.

### A.2  Training and inference details for pertaining and fine-tuning DPViT

**Training Details**. Our approach involves pre-training the Vision Transformer backbone and projection head using the same method described in the iBOT paper [60]. We mostly keep the hyper-parameter settings unchanged without tuning. By default, we use the Vit-Small architecture, which consists of 21 million parameters. The patch size is set to 16 as our default configuration. The student and teacher networks have a shared projection head for the [cls] token output. The projection heads for both networks have an output dimension of 8192. We adopt a linear warm-up strategy for the learning rate over 10 epochs, starting from a base value of 5e-4, and then decaying it to 1e-5 using a cosine schedule. Similarly, the weight decay is decayed using a cosine schedule from 0.04 to 0.4. We employ a multi-crop strategy to improve performance with 2 global crops (224×224) and 10 local crops (96×96). The scale ranges for global and local crops are (0.4, 1.0) and (0.05, 0.4), respectively. Following [60], we use only the local crops for self-distillation with global crops from the same image. Additionally, we apply blockwise masking to the global crops inputted into the student network. The masking ratio is uniformly sampled from [0, 1, 0.5] with a probability of 0.5, and with a probability of 0.5, it is set to 0. Our batch size is 480, with a batch size per GPU of 120. DPViT is pre-trained for 500 epochs for the given training set for all the datasets.

We use the value of $\lambda_{cls} = 1, \lambda_s = 0.5, \lambda_o = 0.5$ for all the datasets. In the case of ImageNet-9, we incorporate the class labels by incorporating a logit head onto the projection heads. This allows us to calculate the cross-entropy loss based on the provided class labels. The explicit utilization of class labels is necessary for the ImageNet-9 dataset because the evaluation involves straightforward classification rather than few-shot learning.

**Fine-tuning Details**. Once the pretraining stage is completed, we proceed to train the model using the supervised contrastive loss, which involves distilling knowledge from [cls] tokens across different views of images (referred to as $\mathcal{L}_{cls}$ in Equation 9 of the main draft). The fine-tuning process is conducted for 50 epochs using the same training data in the given dataset. We maintain the same set of hyperparameters used in the initial pretraining stage without additional tuning.

We use the value of $\lambda_{cls}^{inv} = 1$, and $\lambda_p^{inv} = 0.5$ for all the datasets.

**Inference Details**. For inference purposes, we utilized a feature representation obtained by the [cls] token of the teacher network. We also found concatenating the weighted average pooling of the generated patches with the [cls] token useful in a few-shot evaluation. The weights for the weighted average pooling are determined by taking the average of the attention values of the [cls] token across all heads of the final attention layer.

In the case of ImageNet-9, the logit head is used to infer the class label for the given sample in the test set.

### A.3  Details regarding the multi-head attention modules

The design of our attention layers draws inspiration from the standard self-attention mechanism, commonly known as **qkv** self-attention (SA) [11]. In our implementation, we calculate a weighted sum over all values **v** in the input sequence **z**, where **z** has dimensions of $\mathbb{R}^{N \times D}$. The attention

weights $A_{ij}$ are determined based on the pairwise similarity between two elements of the sequence and their corresponding query $\mathbf{q}^i$ and key $\mathbf{k}^j$ representations.

$$[\mathbf{q}, \mathbf{k}, \mathbf{v}] = \mathbf{z}\mathbf{U}_{qkv} \qquad\qquad \mathbf{U}_{qkv} \in \mathbb{R}^{D \times 3D_h}, \qquad (10)$$

$$A = softmax\left(\mathbf{q}\mathbf{k}^\top / \sqrt{D_h}\right) \qquad\qquad A \in \mathbb{R}^{N \times N}, \qquad (11)$$

$$SA(\mathbf{z}) = A\mathbf{v}. \qquad (12)$$

Multihead self-attention (MSA) is an expansion of the self-attention mechanism, where we perform $k$ parallel self-attention operations, known as "heads," and then combine their outputs through concatenation. In order to maintain consistent computation and the number of parameters when adjusting the value of $k$, the dimension $D_h$ (as defined in Equation 10) is typically set to $D/k$.

$$MSA(\mathbf{z}) = [SA_1(z); SA_2(z); \cdots ; SA_k(z)]\, \mathbf{U}_{msa} \qquad\qquad \mathbf{U}_{msa} \in \mathbb{R}^{k \cdot D_h \times D} \qquad (13)$$

### A.4   Details regarding the power iterative method to compute spectral norm

We follow the power iterative method described in [3] to compute the spectral norm for $(\mathbf{P}^T\mathbf{P} - \mathbf{I})$. Starting with a randomly initialized $v \in \mathbb{R}^n$, we iteratively perform the following procedure a small number of times (2 times by default) :

$$u \leftarrow (\mathbf{P}^T\mathbf{P} - \mathbf{I})v, v \leftarrow (\mathbf{P}^T\mathbf{P} - \mathbf{I})u, \sigma(\mathbf{P}^T\mathbf{P} - \mathbf{I}) \leftarrow \frac{||v||}{||u||}. \qquad (14)$$

The power iterative method reduces computational cost from $\mathcal{O}(n^3)$ to $\mathcal{O}(mn^2)$, which is practically much faster when used with our training procedure.

### A.5   Comparing ViT-S [11] and Concept Transformer (CT) [41] on ImageNet-9

In addition to the findings presented in Section 5.4 (Table 2 in the main draft), we conducted a comparison with vanilla ViT-S pretrained on Imagenet and ConceptTransformers (CT) as well. CT, as described in the study by [41], has a notable limitation in that it relies on attribute supervision for part localization information. This restriction restricts the applicability of CT in scenarios where attribute information is absent, such as in the case of ImageNet-9. To train CT without attributes, we utilized the code provided by the authors and deactivated the attribute loss, allowing CT to be trained without relying on the attribute information [2]. This adjustment significantly decreases the performance of CT but enables a fair comparison with other methods on ImageNet-9. It is worth noting that CT employs the ViT-S backbone pretrained on ImageNet as its default architecture. Moreover, we train ConstNet [54] using the source code provided by the authors [3].

As indicated in Table 8, DPViT demonstrates superior performance compared to both ViT-S pretrained on ImageNet and CT, exhibiting a clear advantage. CT can be seen as a pretrained ViT-S model with the inclusion of part dictionaries, but it experiences a noticeable drop in performance when confronted with the presence of incidental correlations in the image backgrounds (as observed in the low **M-SAME** and **M-RAND** performance in Table 8). This demonstrates that the part learners in general cannot effectively deal with the incidental correlations of backgrounds and are susceptible to varying backgrounds.

### A.6   Ablation study with different values of $K$ and $n_f$ on MiniImageNet

In this analysis, we investigate the impact of varying the number of parts, denoted as $K$, on the MiniImageNet dataset. Specifically, we explore the effects of altering the number of foreground parts, represented by $n_f$, as well as the number of background vectors, which can be calculated as $K - n_f$. Table 9 presents the obtained results, demonstrating the influence of different values of $K$, $n_f$, and $n_b$ on parts, foreground parts, and background parts, respectively.

---

[2]ConceptTransformer [41] - `https://github.com/IBM/concept_transformer`
[3]ConstNet [54] - `https://github.com/mlpc-ucsd/ConstellationNet`

| Method | IN-9L ↑ | Original ↑ | M-SAME ↑ | M-RAND ↑ | BG-GAP ↓ |
|---|---|---|---|---|---|
| **ResNet-50** [53] | 94.6 | 96.3 | 89.9 | 75.6 | 14.3 |
| **WRN-50×2** [53] | 95.2 | 97.2 | 90.6 | 78.0 | 12.6 |
| **ConstNet** [54] | 90.6 | 92.7 | 86.1 | 69.2 | 17.1 |
| **ViT-S pretrained** [11] | 82.5 | 84.9 | 72.2 | 50.3 | 21.9 |
| **ConceptTransformer** [41] | 84.7 | 85.5 | 73.1 | 51.5 | 21.6 |
| **Ours - DPViT** | **96.9** | **98.5** | **93.4** | **87.5** | **5.9** |

Table 8: Performance evaluation on domain shift of varying background and common data corruptions on ImageNet-9. Evaluation metric is Accuracy %.

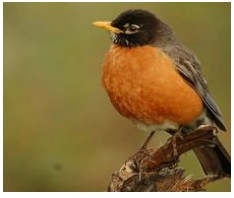 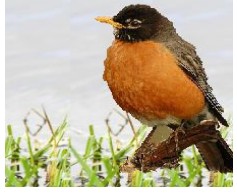 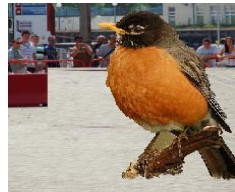

(a) Original    (b) MIXED-SAME    (c) MIXED-RAND

Figure 8: Visualizing the test splits from ImageNet-9 dataset.

| | K=32 | | K=64 | | K=96 | | K=128 | |
|---|---|---|---|---|---|---|---|---|
| **Foreground parts** | **1-shot** | **5-shot** | **1-shot** | **5-shot** | **1-shot** | **5-shot** | **1-shot** | **5-shot** |
| $n_f = K/2$ | $72.2_{\pm0.2}$ | $87.8_{\pm0.4}$ | $72.9_{\pm0.5}$ | $88.1_{\pm0.4}$ | $72.1_{\pm0.2}$ | $88.1_{\pm0.4}$ | $72.1_{\pm0.5}$ | $87.1_{\pm0.5}$ |
| $n_f = 2K/3$ | $72.2_{\pm0.2}$ | $88.1_{\pm0.4}$ | $73.8_{\pm0.5}$ | $89.3_{\pm0.4}$ | $73.1_{\pm0.2}$ | $88.1_{\pm0.4}$ | $72.2_{\pm0.5}$ | $87.4_{\pm0.5}$ |
| $n_f = 4K/3$ | $72.3_{\pm0.2}$ | $88.4_{\pm0.4}$ | $73.4_{\pm0.5}$ | $88.5_{\pm0.4}$ | $73.2_{\pm0.2}$ | $87.9_{\pm0.4}$ | $72.5_{\pm0.5}$ | $87.8_{\pm0.5}$ |

Table 9: Ablation of varying the number of foreground-background vectors, along with part-vectors used. We show the results on the miniImageNet dataset.

Our findings indicate that maintaining $K = 64$ and selecting $n_f = 2K/3$ yields the highest performance. When employing a significantly lower number of part vectors, the model's capacity becomes insufficient, leading to performance degradation. Conversely, employing a larger value of $K$ results in increased computational complexity associated with distance maps, subsequently leading to lower performance.

### A.7    Computational complexity of DPViT

Adding part-dictionaries to MCA layers slightly increases the trainable parameters from $21M$ (ViT-S) to $25M$ (DPViT). It is also possible to share the attention layers, analogous to the Siamese networks, for MSA and MCA, which keeps the number of trainable parameters to $21M$. DPViT results in a similar performance in terms of few-shot accuracy when the attention layers are shared, as shown in Table 10.

### A.8    Stage-1 pertaining comparison with SMKD [30]

Table 11 showcases the few-shot evaluation results of DPViT on the MiniImageNet dataset. In addition, we compare the performance of DPViT with the first-stage performance of SMKD [30]. It is worth noting that both DPViT and SMKD utilize the iBOT [60] pretraining strategy. However, incorporating part-dictionaries, MSA, and MCA layers in DPViT's pretraining phase contributes to its superior performance compared to SMKD.

### A.9    Studying complementary properties of MSA and MCA

Based on Section 5.1 in the main draft, our study focuses on examining the complementary characteristics of MSA and MCA. MSA is designed to be effective for images containing a small number of objects, but it struggles to capture the spatial relationships among multiple objects. In contrast, MCA layers utilize distance maps to learn spatial relationships and prioritize objects without considering

| Setting | 1-shot ↑ | 5-shot ↑ | $\|\mathbf{P}\|_1$ ↓ | $\|\mathbf{P}\mathbf{P}^T - \mathbf{I}\|_1$ ↓ |
|---|---|---|---|---|
| Shared | 73.6 | 89.6 | 0.4 | 0.5 |
| Unshared | 73.8 | 89.8 | 0.3 | 0.5 |

Table 10: **Siamese DPViT**. Sharing MSA and MCA layers and evaluation on MiniImageNet.

| Method | 1-shot ↑ | 5-shot ↑ |
|---|---|---|
| SMKD [30] | 60.93 | 80.38 |
| DPViT | 62.81 | 83.25 |

Table 11: Few-shot performance after $1^{st}$ stage pretrain phase on MiniImageNet.

| Setting | 1-shot ↑ | 5-shot ↑ |
|---|---|---|
| L2-norm | 65.12 | 82.95 |
| L1-norm | 58.73 | 79.51 |
| Cosine | *nan* | *nan* |

Table 12: **Different optimizers for $\mathcal{L}_{mix}$**. Experimenting with different loss terms during pretraining.

| Setting | 1-shot ↑ | 5-shot ↑ |
|---|---|---|
| Gaussian | 65.12 | 82.95 |
| Salt-and-pepper | 62.21 | 82.75 |
| Speckle | 61.60 | 81.50 |

Table 13: Different noise functions for $\delta$ introduced in Equation 4.

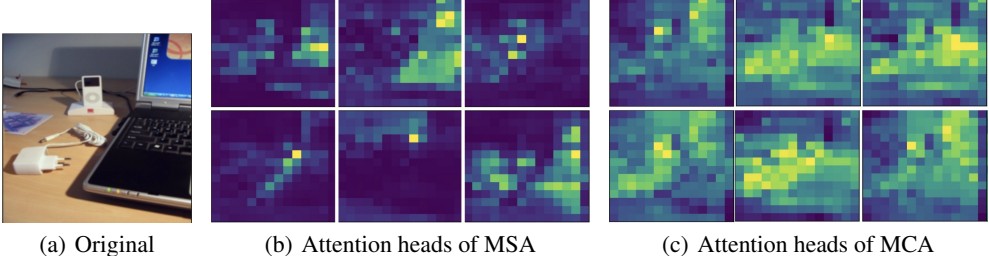

    (a) Original        (b) Attention heads of MSA        (c) Attention heads of MCA

Figure 9: Visualizing the attention heads for MSA and MCA.

their specific classes. In simpler terms, MSA may overlook certain objects that are not crucial for classification, while MCA emphasizes learning spatially similar objects.

Additionally, we present the visualization of attention heads in Figure 9, 10, and 12. The MSA heads excel at identifying objects for classification but may overlook relevant objects with significant spatial context, such as the "charger" in Figure 9 and the "garbage box" in Figure 12. On the other hand, the MCA layers perform well in scenarios involving multiple objects (Figure 9 and 12), but struggle when spatially similar objects are present, as seen with the confusion between the "red grass" and the "fish" in Figure 10.

## A.10   Optimization functions for $\mathcal{L}_{mix}$

The segmentation masks $M_f$ and $M_b$ are composed of binary values. Our selection of the $L_2$-norm for $L_{mix}$ is informed by empirical observations. Additionally, we conducted experiments with alternative loss functions like L1 and cosine distance during the pretraining phase. As shown in Table, while the L1-norm results in lower performance, the cosine distance results in unstable training with training loss reaching +infinity, eventually $NAN$. We achieve the best performance with the L2-norm.

## A.11   Noise functions for $\delta$

We conduct an ablation study on the choice of noise functions used for $\delta$. As shown in Table 13, we ascertain that Gaussian noise is more effective in promoting resilience within the latent codes, while simultaneously preserving the interpretability of parts when compared with salt-and-pepper and speckle noise during the pretraining phase.

## A.12   Visualization foreground parts

We present additional part visualizations for Figure 10(a) and 11(a). These are shown in Figure 12(a) and 12(b).

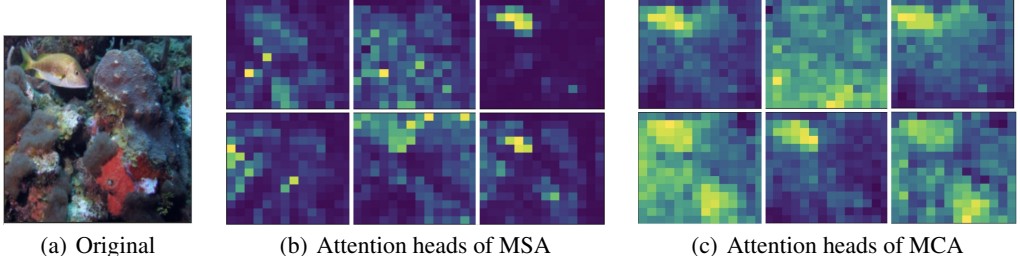

(a) Original     (b) Attention heads of MSA     (c) Attention heads of MCA

Figure 10: Visualizing the attention heads for MSA and MCA.

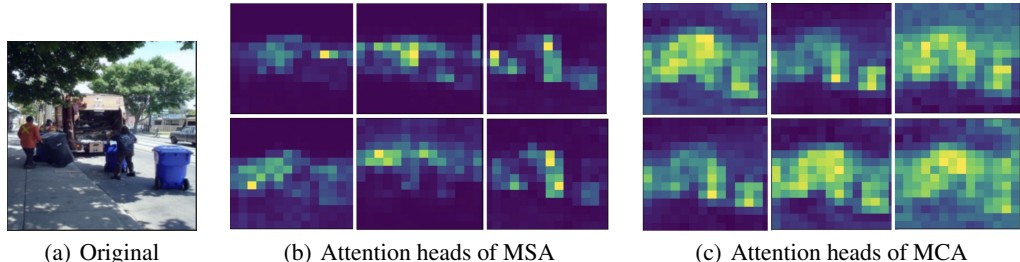

(a) Original     (b) Attention heads of MSA     (c) Attention heads of MCA

Figure 11: Visualizing the attention heads for MSA and MCA.

### A.13 Qualitative comparison of extracted patches with ConstNet [54]

In order to showcase the acquired parts of DPViT and ConstNet, we provide visualizations in Figure 13 and 14. This is achieved by selecting the nearest patches to the parts. Figure 13 illustrates the separation of foreground and background concepts accomplished by our model, whereas Figure 14 exhibits the patches surrounding the learned parts from the ConstNet model.

While DPViT learns to disentangle the foreground patches from the backgrounds, the patches extracted by ConstNet suffer from the entanglement caused due to incidental correlations of backgrounds.

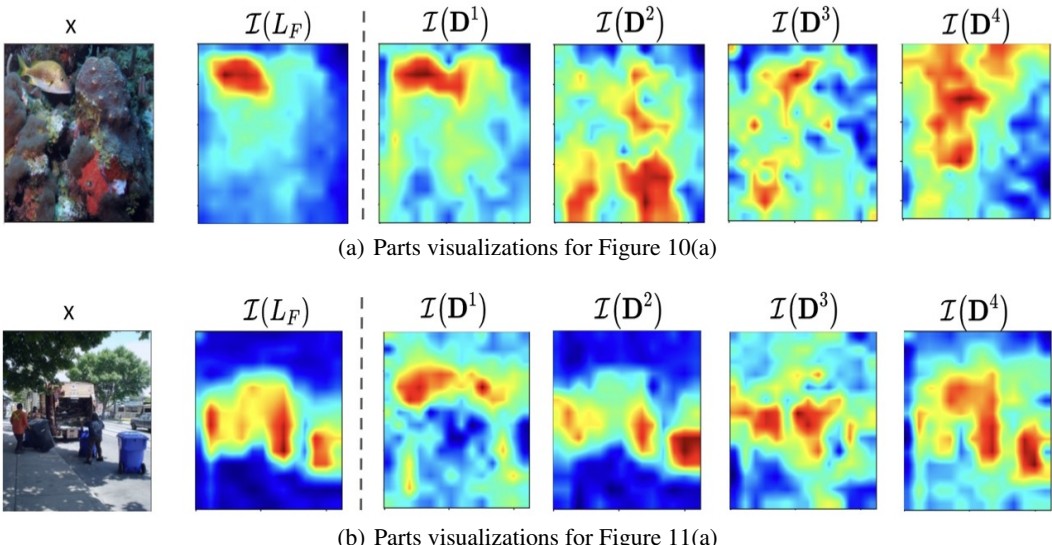

(a) Parts visualizations for Figure 10(a)

(b) Parts visualizations for Figure 11(a)

Figure 12: Visualizing foreground parts learned by DPViT.

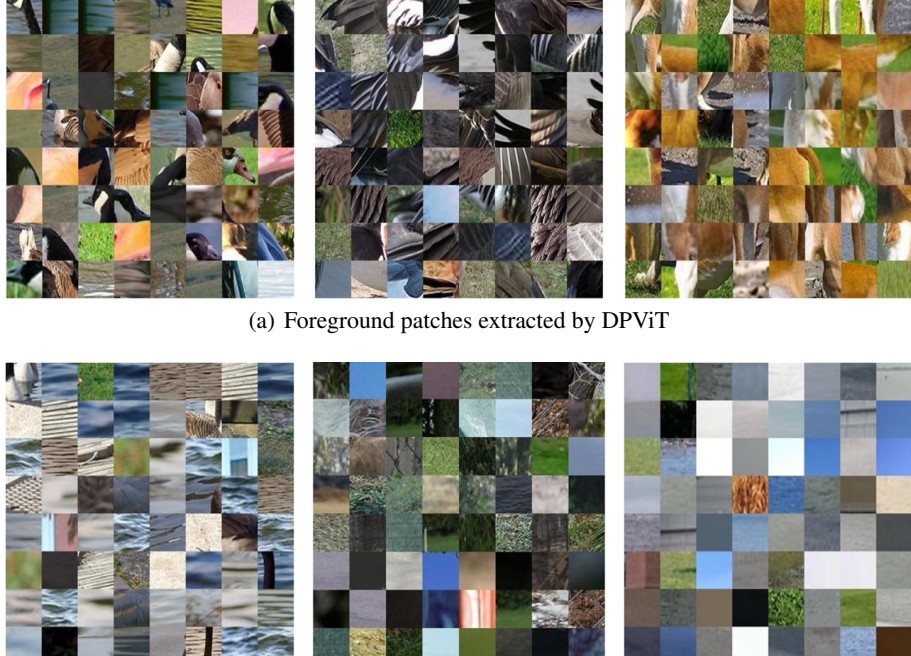

(a) Foreground patches extracted by DPViT

(b) Background patches extracted by DPViT

Figure 13: Visualizing foreground and background patches extracted by DPViT around a random foreground and background part for images from the validation set of MiniImageNet.

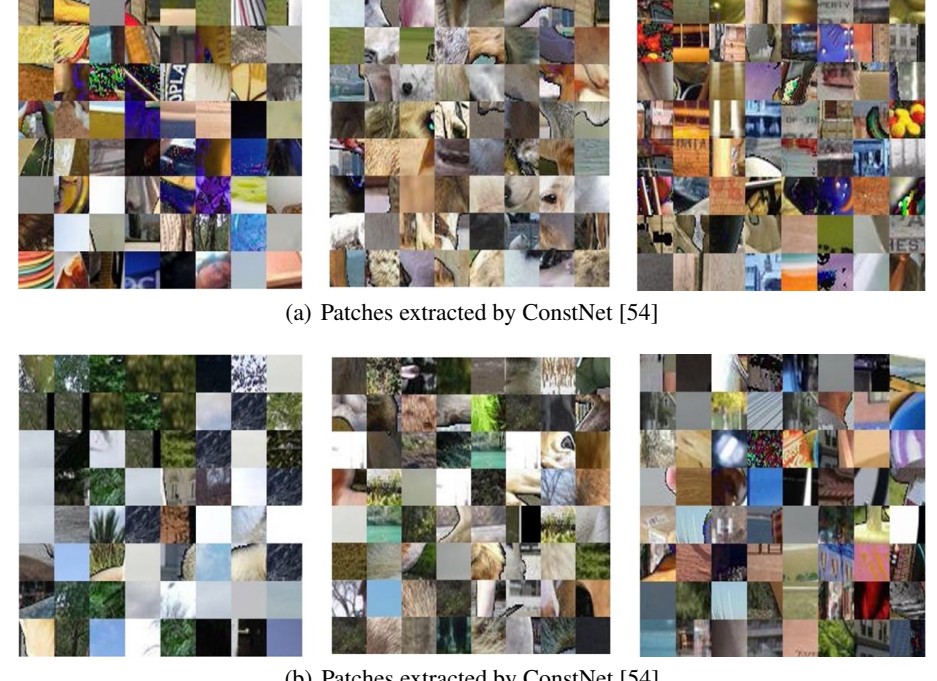

(a) Patches extracted by ConstNet [54]

(b) Patches extracted by ConstNet [54]

Figure 14: Visualizing patches extracted by ConstNet [54] around a random parts for images from the validation set of MiniImageNet.

