# OpenReview forum: "Mitigating the Effect of Incidental Correlations on Part-based Learning"
_NeurIPS.cc/2023/Conference — NeurIPS 2023 poster_

### Official Review · Reviewer_Tpfu · 2023-06-12

**Soundness:** 3 good
**Presentation:** 2 fair
**Contribution:** 3 good
**Rating:** 6
**Confidence:** 5

**Summary:**

This paper builds up a part-based learning algorithm for few-shot classification with components preventing incidental correlations during training. ViT is adopted as the architecture, which facilitates part interpretation by recognizing each image patch as a specific part/foreground/background. Experiments on several few-shot learning benchmarks show the effectiveness of the approach, and an additional experiment on ImageNet-9 verifies that the approach can indeed reduce incidental correlations during training.

**Strengths:**

+ The proposed algorithm achieves interpretable representations to some degree, which is important for few-shot learning.
+ Image background has been shown harmful for few-shot learning at both training and test time, and this paper tackles the background problem at training time in a novel way. The experiments on ImageNet-9 clearly verify that the method can remove incidental correlations.
+ The proposed algorithm achieves SOTA performance on several few-shot learning benchmarks.

**Weaknesses:**

- Missing important ablation studies. (1) To verify the effectiveness of part-based learning, the authors need to show the performance of a vanilla ViT with the loss $L_{cls}$. (2) No ablation studies on the use of $L_{mix}$ and $L_{Q}$. Importantly, the authors do not show the baseline performance using part-based learning with $L_{cls}$ alone during pretraining (especially on few-shot learning benchmarks). Thus the usefulness of removing incidental correlations has not been verified.

- There are confusing parts in the method description. For example, it is not clear if $z_p$ and $z_d$ are iteratively computed for every block of the network or just computed for the feature of the last block. The same confusion goes for distance maps and embedded patches. The architecture plot in Figure 2 does not answer this question (e.g., what's the input to the second block?). Some other confusions are listed in the questions part.

minor:
- A typo in line 188,  it should be $n_f \times F^2 \times C$.




**Questions:**

- Why can Gaussian noise make mixture-of-parts robust to common data distortions? it seems its use is to relax the target $L_{mix}$ to be softer instead.
- What's the exact form of $M_f$ and $M_b$? Are they just binary masks? Why $l_2$-norm is used in $L_{mix}$? Since the target is a mask, it seems cosine distance is more appropriate.
- Please clarify the confusing parts listed in the weaknesses part.

**Limitations:**

No limitations are listed in the paper. I suggest the authors conduct experiments on cross-domain datasets like tieredImageNet->MSCOCO (in Meta-Dataset). Such experiments can show clearer advantages of part-based methods and the removal of incidental correlations.

---

> ### Author Rebuttal · Authors · 2023-08-10
>
> We thank the reviewer for the valuable comments and suggestions. We respond below to each of the concerns/suggestions.
>
> > Missing important ablation studies....
>
> We perform an ablation analysis to investigate the impact of $L_{mix}$ and $L_Q$ during the pretraining phase on the MiniImageNet dataset. Additionally, we provide the baseline results achieved solely through $L_{cls}$, along with the pretraining performance of SMKD [1]:
>
> | Model | 1-shot % | 5-shot % |  $\|P\|_1$ | $\|PP^T - I\|_1$ |
> |--|--|--|--|--|
> SMKD [1] | 60.93 | 80.38 | - |   -
> $L_{cls}$ | 61.24$_{\pm 0.2}$ | 81.12$_{\pm 0.2}$ | 8.41$_{\pm 0.1}$ | 25.82$_{\pm 0.2}$
> $L_{cls}+L_{mix}$ | 62.81$_{\pm 0.1}$ | 83.25$_{\pm 0.2}$ | 8.73$_{\pm 0.2}$ | 24.61$_{\pm 0.1}$
> $L_{cls}+L_{mix}+L_Q$ | 62.15$_{\pm 0.2}$ | 82.95$_{\pm 0.2}$ | 0.35$_{\pm 0.1}$ | 0.56$_{\pm 0.2}$
>
> The above table highlights the relevance of each component during the pretraining.
>
> > There are confusing parts in the method description....
>
> During the architectural design phase, we employ distinct components ($\mathbf{P}$) for each encoder block. Consequently, $z_p$ and $z_d$ are calculated in an iterative manner and subsequently transmitted to the subsequent encoder block. Regarding the computation of $L_{mix}$, we utilize the distance maps (**D**) from the concluding encoder block. While it's feasible to calculate $L_{mix}$ iteratively for each block and then aggregate them for a final $L_{mix}$ computation, our observations indicate that this approach amplifies computational expenses and results in performance deterioration. As a result, we opt to compute $L_{mix}$ using the ultimate encoder block.
> We agree with the reviewer's assessment that the current draft lacks clarity in explaining the method. We intend to rectify this shortcoming in the forthcoming final version by revising the description accordingly.
>
> > Why can Gaussian noise make mixture-of-parts robust to common data distortions?....
>
> Through empirical investigation, we ascertain that Gaussian noise is more effective in promoting resilience within the latent codes, while simultaneously preserving the interpratabilty of parts. In the subsequent table, we delve into the impact of salt-and-pepper and speckle noise during the pretraining phase:
>
>
> | Noise Type | 1-shot % | 5-shot %
> |--|--|--|
> Gaussian | 62.15$_{\pm 0.3}$ | 82.95$_{\pm 0.1}$
> Salt-and-pepper | 62.21$_{\pm 0.2}$ | 82.75$_{\pm 0.3}$
> Speckle | 61.60$_{\pm 0.2}$ | 81.52$_{\pm 0.1}$
>
> Although there isn't a substantial distinction in the few-shot performance during the pretraining stage, we note that the utilization of salt-and-pepper and speckle noise fails to uphold the quality of parts. This, in turn, leads to a decline in the interpretability of part representations, making it advisable to abstain from their use. For qualitative analysis, please refer to figure 1 in the uploaded rebuttal pdf.
>
>
> > What's the exact form of $M_f$ and $M_b$? Are they just binary masks?...
>
> The segmentation masks $M_f$ and $M_b$ are composed of binary values. Our selection of the $L_2$-norm for $L_{mix}$ is informed by empirical observations. Additionally, we conducted experiments with alternative loss functions like L1 and cosine distance during the pretraining phase:
>
>
> | Mix Loss Type | 1-shot % | 5-shot %
> |--|--|--|
> L2-norm | 62.15$_{\pm 0.2}$ | 82.95$_{\pm 0.2}$
> L1-norm | 58.73$_{\pm 0.3}$ | 79.51$_{\pm 0.2}$
> Cosine | (nan loss) | (nan loss)
>
> While L1-norm results in a lower performance, the cosine distance results in unstable training with training loss reaching +infinity, eventually nan. We achieve the best performance with L2-norm.
>
> > No limitations are listed in the paper. I suggest the authors conduct experiments on cross-domain datasets like tieredImageNet-MSCOCO (in Meta-Dataset). Such experiments can show clearer advantages of part-based methods and the removal of incidental correlations.
>
> As suggested by the reviewer, we conduct a study on the cross-domain task. We use the setup from [2] and use MiniImageNet -> CUB evaluation (5-way-5-shot). For this analysis, we refer to the baseline performances reported in Table 9 by [2]. Moreover, we use the official code for ConstNet [3] and SUN (ViT-S) [4] to evaluate cross-domain tasks:
>
>
> | Method | Backbone | MiniImageNet -> CUB
> |--|--|--|
> Neg-Margin* | ResNet-18 | 67.3
> ProtoNet* | ResNet-18 | 67.1
> RelationNet* | ResNet-18 | 57.7
> Baseline* | ResNet-18 | 65.5
> Baseline++* | ResNet-18 | 64.3
> Pos-Margin* | ResNet-18 | 64.9
> MixtFSL [2] | ResNet-18 | 68.7
> Sum-min [2] | ResNet-12 | 67.3
> ConstNet [3] | ResNet-12 | 68.8
> SUN (ViT-S) [4] | ViT-S | 72.1
> **DPViT** | **ViT-S** | **77.1**
> \* results are reported from [2], Table 9.
>
>
> - [1] SMKD: Supervised masked knowledge distillation for few-shot transformers. (CVPR'23)
> - [2] Matching Feature Sets for Few-Shot Image Classification. (CVPR'22)
> - [3] Attentional constellation nets for few-shot learning. (ICLR'21)
> - [4] Self-promoted supervision for few-shot transformer. (ECCV'22)
>
> Moreover, we also include limitations of our work:
> - A constraint within our framework involves relying on a pre-existing foreground extractor. In certain scenarios, such as the classification of tissue lesions for microbiology disease diagnosis, obtaining an existing foreground extractor might not be feasible.
> - At present, DPViT focuses on learning components that are connected to the data, yet it doesn't encompass the connections between these components, like their arrangement and hierarchical combination. Introducing compositional relationships among these components could enhance comprehensibility and facilitate the creation of a part-based model capable of learning relationships among the parts.
>
> > A typo in line 188, it should be $n_f \times F^2\cdot C$
>
> We appreciate your observation of this typographical error. We will make the necessary corrections.

---

> > ### Comment · Reviewer_Tpfu · 2023-08-15
> >
> > Thanks for the authors' response. The additional empirical studies addressed most of my concerns. I raise my score from 5 to 6.

---

> > > ### Author Response · Authors · 2023-08-15
> > >
> > > Thank you for acknowledging our response to your comments. We are happy that most of your concerns have been addressed.

---

### Official Review · Reviewer_QGVs · 2023-07-02

**Soundness:** 3 good
**Presentation:** 3 good
**Contribution:** 3 good
**Rating:** 6
**Confidence:** 3

**Summary:**

The authors propose a strong part-based learning method for few-shot learning. Noticing the incidental correlations between part signals, the method proposed in the paper, named DP-ViT, learned to disentangle foreground and background parts. This is done by introducing a mixture-of-experts formulation of parts. To further encourage high-quality and diverse parts, the DP-ViT further employs a regularization for orthogonality.The method proposed by the authors achieves SOTA on multiple few-shot learning datasets, including MiniImagenet, TieredImageNet, and FC100. With the DP-ViT framework, the authors also illustrate the incidental correlation in imagery data and how DP-ViT improves the performance of models.



**Strengths:**

The authors have made sound improvements to two regularizations on part-based models. For me, these ideas make sense and the authors have demonstrated interpretation of these modules in the experimental sections. Additionally, it is nice to see that the proposed solutions improve the performance of few-shot learning.

**Weaknesses:**

I don't directly work on this topic, and I think the authors have provided adequate analysis and explanation on the few-shot learning part. However, I think this paper could have a more general impact. Please check the questions below.

**Questions:**

1. It looks to me that DP-ViT is a quite general improvement to the ViT architecture. However, to support its full utility, I think an experiment on ImageNet is needed, where DP-ViT can outperform a vanilla ViT or a part-based ViT baseline.

2. I personally would suggest adding an ablation study for the number of parts, K, especially what will happen when K is increased. It is interesting to see that K=64 parts is sufficient for a dataset.

**Limitations:**

I haven't found such a section from the authors. But I tend to think that this work does not have a negative societal impact.

---

> ### Author Rebuttal · Authors · 2023-08-10
>
> We thank the reviewer for the valuable comments and suggestions. We respond below to each of the concerns/suggestions.
>
> > It looks to me that DP-ViT is a quite general improvement to the ViT architecture. However, to support its full utility, I think an experiment on ImageNet is needed, where DP-ViT can outperform a vanilla ViT or a part-based ViT baseline.
>
> As suggested by the reviewer, we conduct an experiment on ImageNet-1K, and compare it with the vanilla ViT-S model.
>
> | Model | Val Acc %|
> |------|------|
> | ViT-S | 54.5 %|
> |**DPViT** | **72.1** %|
>
> Please note that we train both models over 100 epochs using approximately 1 million training samples, employing a batch size of 512. For DPViT, we use the same set of hyperparameters that we used for ImageNet-9 dataset (please refer to Appendix Section 1.2). For ViT-S, we use Adam optmizer with learning rate 0.001 along with cosine scheduler (the code is adopted from **timm** library). Subsequently, we evaluate the models using the provided validation set consisting of 50,000 samples. Ideally, DPViT could undergo pretraining on an expansive dataset like ImageNet-21K or JFT300M, followed by fine-tuning on ImageNet-1K. Regrettably, due to limitations in time and resources, we were unable to carry out such extensive pretraining for both models.
>
> > I personally would suggest adding an ablation study for the number of parts, K, especially what will happen when K is increased. It is interesting to see that K=64 parts is sufficient for a dataset.
>
> We have reported an ablation with a varying number of parts K on MiniImageNet dataset in Appendix Section 1.6 (Table 2):
>
> | | K=32 | K=64 | K=96 | K=128 |
> |--|--|--|--|--|
> |$n_f$ | 1-s/5-s | 1-s/5-s | 1-s/5-s |1-s/5-s|
> | $n_f=K/2$ | $72.2_{\pm 0.2}$ / $87.8_{\pm 0.4}$ | $72.9_{\pm 0.5}$ / $88.1_{\pm 0.4}$| $72.1_{\pm 0.2}$ / $88.1_{\pm 0.4}$ | $72.1_{\pm 0.5}$/ $87.1_{\pm 0.5}$
> | $n_f=2K/3$ | $72.2_{\pm 0.2}$/ $88.1_{\pm 0.4}$ |$73.8_{\pm 0.5}$ / $89.3_{\pm 0.4}$ | $73.1_{\pm 0.2}$ / $88.1_{\pm 0.4}$ | $72.2_{\pm 0.5}$ / $87.4_{\pm 0.5}$
> |$n_f = 4K/3$ | $72.3_{\pm 0.2}$/ $88.4_{\pm 0.4}$ | $73.4_{\pm 0.5}$/ $88.5_{\pm 0.4}$ |$73.2_{\pm 0.2}$/ $87.9_{\pm 0.4}$ | $72.5_{\pm 0.5}$/ $87.8_{\pm 0.5}$
>
> We observed that increasing $K$ beyond a certain threshold degrades the performance as computational complexity increases.
>
>
> > Limitations: I haven't found such a section from the authors. But I tend to think that this work does not have a negative societal impact.
>
> Limitations of our work:
> - A constraint within our framework involves relying on a pre-existing foreground extractor. In certain scenarios, such as the classification of tissue lesions for microbiology disease diagnosis, obtaining an existing foreground extractor might not be feasible.
> - At present, DPViT focuses on learning components that are connected to the data, yet it doesn't encompass the connections between these components, like their arrangement and hierarchical combination. Introducing compositional relationships among these components could enhance comprehensibility and facilitate the creation of a part-based model capable of learning relationships among the parts.

---

> > ### Comment · Reviewer_QGVs · 2023-08-11
> >
> > The authors addressed most of my concerns. I also read the reviews from fellow reviewers, especially the reviewers with "rejection" ratings, and I temporarily keep my ratings.
> >
> > Thanks for pointing out the ablations for the number of parts (K), it makes sense to me.
> >
> > A follow-up: I appreciate adding the first experiment of comparing with ViT-S. However, the performance of ViT-S is not quite right (even under the 100 epochs setting). I have run this before myself and the accuracy is around 75%. I suggest using the [Deit](https://github.com/facebookresearch/deit) repository for training. Nonetheless, I understand that the authors have limited resources, and this is not a claimed contribution in the paper. So this won't affect my ratings. Please remember to figure this out for camera-ready if you get accepted in the future.

---

> > > ### Author Response · Authors · 2023-08-11
> > >
> > > We appreciate your reply. It's reassuring to see that many of the issues you raised have been taken care of. Following your recommendation, we intend to focus on improving the ViT-S outcomes in the upcoming version, particularly by utilizing the promising Deit framework.
> > > Please let us know if you need any other information from our side.

---

> > > > ### Author Response · Authors · 2023-08-17
> > > > **A gentle reminder (Reviewer QGVs)**
> > > >
> > > > We have responded to all inquiries raised by Reviewer gyRR (who initially gave a rejection rating). It appears that all the issues highlighted by Reviewer gyRR have been appropriately tackled, leading to their willingness to raise the score to 6. We kindly request you to review our discussion with Reviewer gyRR and contemplate the possibility of adjusting your score.
> > > >
> > > > Once more, we express our genuine gratitude for your dedication throughout the review procedure and subsequent discussions.

---

> > > > > ### Author Response · Authors · 2023-08-20
> > > > > **Final kind reminder (Reviewer QGVs)**
> > > > >
> > > > > Since the author-reviewer discussion period concludes on Monday, we're interested in confirming whether our response adequately addressed the reviewers' paper-related concerns. Please inform us if you require additional clarification, and we will make every effort to provide a response by tomorrow.
> > > > >
> > > > > Once again, we appreciate your feedback and the time you've dedicated to the review.

---

### Official Review · Reviewer_gyRR · 2023-07-05

**Soundness:** 2 fair
**Presentation:** 3 good
**Contribution:** 3 good
**Rating:** 6
**Confidence:** 3

**Summary:**

The paper addresses the impact of incidental correlations on part-learning and proposes several regularization methods to mitigate this. The first regularization separates foreground and background to guide the part-based learning towards relevant input regions. To this end, the paper proposes a mixture of parts formulation to collect latent codes into masks and supervise them (weakly) with the pretrained/unsupervised foreground extractor. Then, the second regularization is employed to impose invariance to background variation. Additionally, sparsity and orthogonality regularization is used to refine part representations.

**Strengths:**

1. The paper tackles an important problem. Part-based methods are becoming more and more valuable nowadays due to increased interpretability
2. The paper points out the failure mode of part-based learning
3. The text is nicely written and easy to follow. Figures nicely highlight most relevant findings.

**Weaknesses:**

1. The choice of sparse and spectral norms is not well explained in my opinion. I understand the need for sparsity in the image space when training part-based models, but in the paper, sparsity is enforced on the matrix of part representations instead. What is the physical meaning of this? Also, why is the spectral norm chosen to optimize for orthogonality? It is not evident straight away why would it be a better alternative to just L2 / Frobenius.

2. Why the background model is needed as a separate entity? Is modeling just the foreground (the background is then all the rest of the image naturally) not sufficient? When the background is modeled explicitly as a mixture of parts, then the part-model should learn to encode all background patches as well, right? This looks like an unnecessary complication.

3. The whole framework consists of 3 regularizers, each with its weighting. This introduces more hyper-parameters to the model and increases resources needed to optimize the performance. With this, it is not clear if the (rather limited) performance improvement provided by the method is worth the increased demand for hyper-parameter optimization.

4. Most of the methods in experimental evaluation are built on top of different backbones. It makes it harder to disentangle improvement from regularization and improvement from the backbone. It would make sense to use the model trained with L_{cls} only as a baseline for comparison in Table 1 and Table 2.

UPD: Authors addressed most of my concerns in the rebuttal.

**Questions:**

I suggest authors address weaknesses for the rebuttal.

**Limitations:**

I suggest authors elaborate on hyper-parameter optimization for the proposed framework.

---

> ### Author Rebuttal · Authors · 2023-08-10
>
> We thank the reviewer for the valuable comments and suggestions. We respond below to each of the concerns/suggestions.
>
> > The choice of sparse and spectral norms is not well explained in my opinion. I understand the need for sparsity in the image space when training part-based models, but in the paper, sparsity is enforced on the matrix of part representations instead. What is the physical meaning of this? ...
>
> We have explained the need of sparse and orthogonal norm in the context of our work in L57-L65 : “*Sparsity ensures that only a few parts are responsible for a given image, as images comprise a small subset of parts. Conversely, diversity in part representations prevents the parts from converging into a single representation and facilitates each part’s learning of a unique data representation*”. The need of sparsity is to select a subset of parts from the part-matrix for a given sample. Similarly, the need of orthogonality is to impose diversity in the foreground/background parts.
>
> Moreover, the choice of spectral norm is explained in L200-L213 : “ *One solution is to enforce orthogonality on the matrix $\mathbf{P}^{m\times n}$ by minimizing $||\mathbf{P}^T\mathbf{P} - \mathbf{I}||$. However, the solution will result in a biased estimate as $m<n$; that is, the number of parts ($K$) is always less than the dimensionality of parts ($F^2\cdot C$). In our experiments, we observed that increasing $K$ beyond a certain threshold degrades the performance as computational complexity increases. (Please refer to our Appendix section 1.6 (Table 2) for experiments on the different values of $K$). To minimize the degeneration of parts, we design our quality assurance regularization by minimizing the spectral norm of $\mathbf{P}^T\mathbf{P} - \mathbf{I}$, and by adding $L_1$ sparse penalty on the part-matrix $\mathbf{P}$. The spectral norm of $\mathbf{P}^T\mathbf{P} - \mathbf{I}$ has been shown to work with over-complete ($m<n$) and under-complete matrices ($m\geq n$)* [1].”
> We structure our power iterative spectral norm methodology based on the framework presented in [1]. It's important to emphasize that the study conducted in [1] concentrates on ensuring orthogonality across all weights within the neural network. Conversely, in our approach, we employ spectral norm to introduce diversity among the matrices associated with foreground and background components. For a comprehensive verification of the spectral norm concept, please refer to [1]"Can we gain more from orthogonality regularizations in training deep networks?" (NeurIPS 2018).
>
>
> >Why the background model is needed as a separate entity? Is modeling just the foreground (the background is then all the rest of the image naturally) not sufficient? ...
>
> The background parts provides essential contextual information during training, which is cruical for learning interpretable part representations. To verify this, we conduct an ablation study where n_f = K, and n_b = 0. Furthermore, we switch off the background influence on $L_{mix}$ by modifying the Equation 5 and 6 as:
>
> $\mathcal{L}_{mix} = || \mathcal{I}(L_F) - \mathcal{M}_f||_2$
>
> $\mathcal{L}_{Q} (\lambda_s, \lambda_o) = \lambda_s ||\mathbf{P} ||_1 + \lambda_o \Big[\sigma \big(\mathbf{P_F} \cdot \mathbf{P_F}^{T} - \mathbf{I} \big) \Big] $
>
> | Design |1-shot %| 5-shot %  |  $\|P\|_1$ | $\|PP^T - I\|_1$ |
> | -------- | -------- | -------- | -- | -- |
> | Foreground |72.81$_{\pm{0.15}}$ |88.21$_{\pm{0.18}}$ | 0.35$_{\pm{0.21}}$| 0.99$_{\pm{0.11}}$ |
> | Foreground+Background |73.05$_{\pm{0.15}}$ | 88.56$_{\pm{0.18}}$ | 0.32$_{\pm{0.21}}$| 0.32$_{\pm{0.17}}$ |
>
> The foreground-only model exhibit comparable few-shot performance to foreground+background model, but the acquired components lack diversity, resulting in reduced interpretability. This is evident from the higher orthogonal-norm value observed in the case of the foreground-only model.
>
> >The whole framework consists of 3 regularizers, each with its weighting. This introduces more hyper-parameters to the model and increases resources needed to optimize the performance. ...
>
> Although our framework incorporates multiple hyperparameters, extensive hyperparameter tuning is not required. As outlined in Appendix Section 1.2, we maintain consistency in most hyperparameters across all employed datasets. Specifically, we set $\lambda_{cls}$ = 1, $\lambda_{s}$ = 0.1, $\lambda_o$ = 0.1, $\lambda_{cls}^{inv}$ = 1, and $\lambda_p^{inv}$ = 0.5 for all experiments conducted on MiniImagenet, TieredImageNet, FC100, Imagenet-9, and Imagenet-1K datasets. For a more comprehensive understanding of our approach to hyperparameter optimization, kindly refer to Appendix Section 1.2.
>
> >Most of the methods in experimental evaluation are built on top of different backbones. It makes it harder to disentangle improvement from regularization and improvement from the backbone. It would make sense to use the model trained with L_{cls} only as a baseline for comparison in Table 1 and Table 2.
>
> For a better evaluation of our work, we conduct experiments with the $L_{cls}$ baseline. As suggested by the reviewer, we compare $L_{cls}$ model with DPViT in Table 1 and Table 2 as follows:
>
> Table1:
> | | MiniImageNet | TieredImageNet  | FC100  |
> | -------- | -------- | -------- | -------- |
> | Model |1-shot/5-shot | 1-shot/5-shot | 1-shot/5-shot|
> | $L_{cls}$ |$72.15_{\pm{0.20}}$/87.61$_{\pm{0.15}}$ | $78.03_{\pm{0.19}}$/89.08$_{\pm{0.19}}$ | $48.92_{\pm{0.13}}$/67.75$_{\pm{0.15}}$|
> | DPViT |$73.81_{\pm{0.17}}$/89.85$_{\pm{0.18}}$ | $79.32_{\pm{0.19}}$/91.92$_{\pm{0.20}}$ | $50.75_{\pm{0.20}}$/68.80$_{\pm{0.15}}$|
>
> Table 2:
> |	Method | IN-9L | Org	|M-SAME	|M-RAND |BG-GAP|
> |--------| -------- | -------- | -------- | -------- | -------- |
> |$L_{cls}$ | 95.1$_{\pm{0.21}}$ | 97.2$_{\pm{0.25}}$ | 91.5$_{\pm{0.19}}$ | 81.7$_{\pm{0.15}}$ | 9.2$_{\pm 0.2}$ |
> |DPViT | 96.9$_{\pm{0.15}}$ | 98.5$_{\pm{0.20}}$ | 93.4$_{\pm{0.19}}$ | 87.5$_{\pm{0.23}}$ | 5.9$_{\pm 0.2}$ |

---

> > ### Comment · Reviewer_gyRR · 2023-08-10
> >
> > Authors addressed most of my concerns.
> >
> > Can authors elaborate more on the sparsity part?
> > “Sparsity ensures that only a few parts are responsible for a given image, as images comprise a small subset of parts." <- This I understand, but in the paper the sparsity is enforced on the whole matrix of part representations, meaning it does not regularize the number of parts that an image is comprised of, but it rather regularizes the sparsity of individual part representations, which is not the same.

---

> > > ### Author Response · Authors · 2023-08-11
> > >
> > > Thank you for acknowledging our response. We are happy that most of your concerns have been addressed.
> > >
> > > In our implementation, we employ a modified form of a sparse constraint. We utilize the part matrix $P$ consisting of $K$ parts, structured as $K\times F^2 \cdot C$, where we arrange data along the rows and compress the columns (averaging each row). This operation results in a one-dimensional row vector with a dimension of ($K\times 1$) where each dimension indicates the contribution of the corresponding part vector. Ultimately, we apply our sparse constraint to the column vector  ($1 \times K$).
> > >
> > > Hope this clarifies. If there is any more information we can provide, please let us know.

---

> > > > ### Author Response · Authors · 2023-08-16
> > > > **A gentle reminder (Reviewer gyRR)**
> > > >
> > > > Thank you again for your thoughtful review. Does our response help address your concerns? We would appreciate the opportunity to engage further if needed.

---

> > > > > ### Comment · Reviewer_gyRR · 2023-08-16
> > > > >
> > > > > I thank authors for the response. The rebuttal addresses my concerns, given rebuttal experiments and clarifications are added to the main paper. I thus raise my score.

---

> > > > > > ### Author Response · Authors · 2023-08-17
> > > > > > **A gentle reminder regarding score update on OpenReview**
> > > > > >
> > > > > > We are happy that all your concerns have been addressed. We sincerely appreciate the thorough discussion about our methodology which brought out important insights that might have otherwise been overlooked. We will ensure that all the necessary details and ablations are included in the final version of our paper.
> > > > > >
> > > > > > On a separate note, we observed that the updated score hasn't been reflected in the OpenReview platform yet (and is still 4). Therefore, we kindly request you to consider revising the score accordingly. Once more, we express our gratitude for the time you've dedicated to these thoughtful discussions.

---

### Official Review · Reviewer_DtGy · 2023-07-05

**Soundness:** 4 excellent
**Presentation:** 3 good
**Contribution:** 3 good
**Rating:** 6
**Confidence:** 4

**Summary:**

The paper proposes to produce a more robust and interpretable part-based learning method for image classification problems (though, probably extensible to other tasks). The main contributions include 1) the Disentanglement of foreground/background regions via a weakly supervised loss and 2) the Use of sparse and orthogonality constraints to discourage degenerate solutions 3) Finetuning using invariance constraint to ensure predictions are invariant for the learned background parts. The experiments show competitive results compared to recent methods and provide (arguably) more interpretable results.

**Strengths:**

[S1] Clearly motivated and technically sound: The paper addresses a well-known problem of spurious correlations in neural networks. While there are multiple types of spurious correlations, the paper only focuses on spurious correlations stemming from foreground and background regions. While that makes it more limited, it also provides clearer motivation, and the solution proposed can also address the specific problem better.  Indeed, the proposed solutions appear technically sound and well-designed to discourage correlations which are also well corroborated by the results.

[S2] Adequate details/ablations: There are many parts to the proposed method, but the paper has done a reasonably good job with ablations and discussion of the effects of different components.

**Weaknesses:**

[W1] Some details are slightly unclear: The pretraining portion is mostly quite clear but I have some clarity issues with the finetuning (distillation) portion of the proposed method. I am not sure I fully grasp how Eqn 8 part 2, ensures the fg code captures details in the absence of background. Part 1 is clearer but still needs a bit more details (i.e., if the goal is to make predictions invariant of background, why distill from a teacher model that has access to both? I assume that the choice is made based on empirical results; e.g., fg only model might not converge well), but this needs to be discussed in more detail.

[W2] Effects of error propagation from foreground/background "weak supervision": The existing solution requires access to a model that can segment/divide images into fg and bg during training. While I agree that such models are easy to use and are widely available, the paper is unclear about what the effects of the quality of the underlying model are on the proposed method. The quality of fg/bg themselves might have issues with correlations. This could perhaps be simulated by using corruption techniques to see how the quality of foreground/background segmentation affects proposed methods and also get an idea of how this method might fare with newer/future methods for fg/bg segmentation. L333 studies the availability of masks but not whether a noisy/incorrect mask might affect the model.

**Questions:**

- Why is the orthogonality constraint applied to the whole matrix rather than foreground/background separately? Is this done for practical reasons (i.e., it would be difficult to implement) or is there a motivation behind this choice?

- Please see W1.

**Limitations:**

The paper aims to reduce spurious correlations that can help mitigate some biases/lack of generalization in neural networks. I do not see an explicit potential for negative societal impact. The limitation section is missing and could include technical limitations/assumptions behind the work (Does not address intra-fg spurious correlations, requires access to masks, etc..), but is not a deal-breaker.

---

> ### Author Rebuttal · Authors · 2023-08-10
>
> We thank the reviewer for the valuable comments and suggestions. We respond below to each of the concerns/suggestions.
>
> > [W1] Some details are slightly unclear: The pretraining portion is mostly quite clear but I have some clarity issues with the finetuning (distillation) portion of the proposed method. I am not sure I fully grasp how Eqn 8 part 2, ensures the fg code captures details in the absence of background. Part 1 is clearer but still needs a bit more details (i.e., if the goal is to make predictions invariant of background, why distill from a teacher model that has access to both? I assume that the choice is made based on empirical results; e.g., fg only model might not converge well), but this needs to be discussed in more detail.
>
> In the process of fine-tuning, the initial component of Equation 8 (referred to as $L_{cls}^{inv}$) guarantees that the student model acquires pertinent foreground insights from a teacher model equipped with access to both foreground and background information. This design decision further guarantees the incorporation of pertinent background insights during the learning of representations. As recommended, we carry out an empirical investigation to examine the impact of knowledge distillation from a teacher model possessing solely foreground information:
>
> |Architecture	|1-shot %	| 5-shot %|
> |--------| -------- | -------- |
> |Distill only foreground	|73.69$_{\pm{0.23}}$	| 88.51$_{\pm{0.17}}$	|
> |Distill foreground + background	|73.81$_{\pm{0.21}}$	|89.85$_{\pm{0.20}}$|
>
> As demonstrated in the table above, the teacher model equipped with the ability to distill knowledge from both foreground and background information exhibits slightly superior performance compared to the model that solely accesses foreground-related information.
>
> During the finetuning process, we abstain from employing the mixture loss due to the utilization of learned latent codes for generating foreground masks. The incorporation of $L_p^{inv}$ essentially confines the foreground elements within the foreground space. Alternatively, another approach would involve utilizing the foreground masks to calculate the mixture loss (akin to the pretraining stage). If this route is taken, the inclusion of the $L_p^{inv}$ loss becomes unnecessary when fine-tuning. Through empirical testing, we have confirmed that both of these design options yield comparable performance outcomes.
>
> > [W2] Effects of error propagation from foreground/background "weak supervision": The existing solution requires access to a model that can segment/divide images into fg and bg during training. While I agree that such models are easy to use and are widely available, the paper is unclear about what the effects of the quality of the underlying model are on the proposed method. The quality of fg/bg themselves might have issues with correlations. This could perhaps be simulated by using corruption techniques to see how the quality of foreground/background segmentation affects proposed methods and also get an idea of how this method might fare with newer/future methods for fg/bg segmentation. L333 studies the availability of masks but not whether a noisy/incorrect mask might affect the model.
>
> We agree with the reviewer that studying the quality of foreground/background segmentation could be simulated by adding corruptions to the ground truth masks. We conduct an ablations studying the effect of corrupting the ground truth masks using the Gaussian noise with a manual corruption strength ($\lambda_c$) :
>
> $M_f = \lambda_c \delta + M_f; M_b = \lambda_c \delta + M_b; \delta \sim N(0,1)$.
>
> | | $\lambda_c$=0.1 | $\lambda_c$=0.5  | $\lambda_c$=1.0  |
> | -------- | -------- | -------- | -------- |
> | Method |1-shot/5-shot | 1-shot/5-shot | 1-shot/5-shot|
> | DPViT | $73.05_{\pm{0.15}}$  / 88.56$_{\pm{0.18}}$ | $72.83_{\pm{0.20}}$/ 87.92$_{\pm{0.19}}$ | $72.12_{\pm{0.17}}$/ 87.51$_{\pm{0.15}}$ |
>
> As depicted in the Table above, DPViT displays a degree of robustness against minor to moderate corruptions in segmentation masks (<=0.5). However, once this threshold is surpassed, the interpretability of DPViT experiences a marked decline. We substantiated this observation through qualitative analysis. Please refer to Figure 2 and 3 in the uploaded rebuttal pdf for qualitative analysis.
>
> > Why is the orthogonality constraint applied to the whole matrix rather than foreground/background separately? Is this done for practical reasons (i.e., it would be difficult to implement) or is there a motivation behind this choice?
>
> We would like to clarify that the orthogonality is applied to foreground and background parts separately rather than the whole matrix (as indicated by Equation 6):
>
> $\mathcal{L}_{Q} (\lambda_s, \lambda_o) = \lambda_s ||\mathbf{P} ||_1 + \lambda_o \Big[\sigma \big(\mathbf{P_F} \cdot \mathbf{P_F}^{T} - \mathbf{I} \big)+ \sigma \big( \mathbf{P_B} \cdot \mathbf{P_B}^{T} - \mathbf{I}\big) \Big] $
>
> > Limitations:
> The paper aims to reduce spurious correlations that can help mitigate some biases/lack of generalization in neural networks. I do not see an explicit potential for negative societal impact. The limitation section is missing and could include technical limitations/assumptions behind the work, but is not a deal-breaker.
>
> Limitations of our work:
> - A constraint within our framework involves relying on a pre-existing foreground extractor. In certain scenarios, such as the classification of tissue lesions for microbiology disease diagnosis, obtaining an existing foreground extractor might not be feasible.
> - At present, DPViT focuses on learning components that are connected to the data, yet it doesn't encompass the connections between these components, like their arrangement and hierarchical combination. Introducing compositional relationships among these components could enhance comprehensibility and facilitate the creation of a part-based model capable of learning relationships among the parts.

---

> > ### Author Response · Authors · 2023-08-16
> > **A gentle reminder (Reviewer DtGy)**
> >
> > Thank you again for your thoughtful review. Does our response help address your concerns? We would appreciate the opportunity to engage further if needed.

---

> > > ### Author Response · Authors · 2023-08-20
> > > **Final kind reminder (Reviewer DtGy)**
> > >
> > > Since the author-reviewer discussion period concludes on Monday, we're interested in confirming whether our response adequately addressed the reviewers' paper-related concerns. Please inform us if you require additional clarification, and we will make every effort to provide a response by tomorrow.
> > >
> > > Once again, we appreciate your feedback and the time you've dedicated to the review.

---

### Author Rebuttal · Authors · 2023-08-10

We thank all reviewers for their positive feedback: proposed method tackles an important problem in a novel way with increased interpretability in representations [gyRR, Tpfu]; is clearly motivated and is technically sound [DtGy, QGVs]; achieves state-of-the-art performance with improvements over existing methods [QGVs, Tpfu]; has adepquate details/ablations [DtGy]; is nicely written and easy to follow [gyRR]. We respond to the comments of each reviewer individually below.

---

### Author Response · Authors · 2023-08-21
**General Response to All**

Dear AC and reviewers,

We wish to express our gratitude for your endeavors to participate in the review process.  Our utmost efforts have been dedicated to resolving the concerns articulated by all reviewers. Following extensive deliberation, those reviewers who initially expressed reservations (gyRR and Tpfu) have revised their evaluations to indicate acceptance. Specifically, we have incorporated the following experiments/ablations/clarification as suggested by gyRR and Tpfu:
- Clarification for using sparse and spectral norms in the context of our work.
- Comparison with $L_{cls}$ baseline on all the datasets (Table 1 and Table 2 of our main draft).
- Conceptual reason (backed by empirical results) for using foreground+background information.
- Ablation study of each component of our objective function during pretraining: $L_{cls}$, $L_{mix}$, and $L_{Q}$.
- Clarification on our proposed architecture and methodology.
- Empirical results for using specific forms of noise and mixture objective.
- Results on cross-domain task: MiniImageNet -> CUB, and comparison with strong baselines and SOTA FSL methods.

We have diligently addressed suggestions/clarifications from (DtGy and QGVs), who initially gave us a positive response, and believe we have only strengthened their conviction through our responses.
Specifically, we have incorporated the following experiments/ablations/clarification as suggested by gyRR and Tpfu:
- Conceptual justification (backed by empirical results) for distilling knowledge from the foreground+background model instead of the foreground-only model.
- Conducting an ablation study employing corrupted masks to investigate the resilience of DPViT in the face of low-quality ground-truth masks.
- Limitations of our work that were not addressed in our initial submission.
- Clarification on a varying number of parts K on the MiniImageNet dataset.
- Conducting experiments on the ImageNet dataset and contrasting them with the performance of the standard ViT-S model. In response to the query raised by reviewer QGVs, we intend to include experiments utilizing the Deit framework on ImageNet in the final version.

While we could not engage further with some reviewers and would have loved to hear their subsequent comments, we hope you all would acknowledge our sincere efforts to address all the concerns raised by the reviewers. Needless to say, we would be happy to provide any more information if required and would incorporate these discussions in the final version of the manuscript.

Sincerely,

Authors of 12478

---

### Decision · Program_Chairs · 2023-09-21

**Decision:**

Accept (poster)

**Comment:**

All reviewers found the proposed method to be sound and effective and the results promising. The rebuttal successfully addressed reviewer comments and all reviewers recommend acceptance. The authors are encouraged to improve the final paper version by following reviewer recommendations.